# Understanding the Interplay between Parametric and Contextual Knowledge for Large Language Models

## Abstract

Large language models (LLMs) encode vast amounts of knowledge during pre-training (parametric knowledge, or `PK`) and can further be enhanced by incorporating contextual knowledge (`CK`). Can LLMs effectively integrate their internal `PK` with external `CK` to solve complex problems? In this paper, we investigate the dynamic interaction between `PK` and `CK`, categorizing their relationships into four types: *Supportive, Complementary, Conflicting*, and *Irrelevant*. To support this investigation, we introduce ECHOQA, a benchmark spanning scientific, factual, and commonsense knowledge. Our results show that LLMs tend to suppress their `PK` when contextual information is available, even when it is complementary or irrelevant. While tailored instructions can encourage LLMs to rely more on their `PK`, they still struggle to fully leverage it. These findings reveal a key vulnerability in LLMs, raising concerns about their reliability in knowledge-intensive tasks.

## 1 Introduction

> " *If a man keeps cherishing his old knowledge, so as continually to be acquiring new.* "
> — *Confucius*

Large language models (LLMs) (Brown et al., 2020; Ouyang et al., 2022; Touvron et al., 2023; Zeng et al., 2023; OpenAI, 2024) capture a substantial amount of knowledge through pre-training on large-scale corpus, referred to as *parametric knowledge* (`PK`). When applied to problem-solving, LLMs often rely on additional information provided as context, known as *contextual knowledge* (`CK`)[1]. A critical area that remains underexplored is how effectively LLMs can integrate `PK` and `CK` to solve problems. This ability is crucial for human-like intelligence, where we effortlessly draw upon stored knowledge to adapt to new environments, reflecting our capacity to *perceive, recall, and integrate* information when encountering new stimuli (Gibson, 1988). Effective collaboration between `PK` and `CK` is also vital for real-world applications, such as retrieval-augmented generation (RAG) (Schick et al., 2023; Trivedi et al., 2023; An et al., 2024; Gutiérrez et al., 2024).

However, *we find that for LLMs, an effective interaction between `PK` and `CK` can be challenging.* For example, we observe a counter-intuitive phenomenon that when supplementing an LLM with more task-relevant knowledge in the context (`CK`), sometimes the performance degrades because the LLM fails to effectively utilize the parametric knowledge (`PK`). Examples are shown in Figure 1. This catastrophic test time forgetting greatly limits LLMs capability to effectively solve user queries when relevant context is provided. In this paper, our objective is to thoroughly investigate *the intricate dynamics between `PK` and `CK`*, leading to a better understanding of how well LLMs integrate and prioritize these sources of knowledge. Previous works separately study when `CK` is conflicting or irrelevant to `PK`, raising concerns about the vulnerability (Wang et al., 2023b; Xie et al., 2024; Wu et al., 2024). However, they fail to answer more fine-grained questions such as: To what extent are LLMs aware of `PK`, in the presence of `CK`? How well can LLMs leverage `PK` given `CK`? What factors can affect such ability?

To perform a systematic study, we formulate *reasoning types* based on various relationships between `CK` and `PK` —*Supportive, Complementary, Conflicting* and *Irrelevant*, illustrated by examples in Figure 1 and Table 1. Due to the lack of appropriate datasets to examine this ability, we introduce

---

[1]For simplicity, `PK` and `CK` denote parametric and contextual knowledge throughout this paper.

Figure 1: Our benchmark ECHOQA, accessing LLMs ability to echo their parametric knowledge (PK) when contextual knowledge (CK) is present. We firstly question LLMs to obtain PK and discard knowledge they cannot answer. Then, we construct CK by various *reasoning types* (Table 1). Next, we question LLMs given CK. Exemplar result is by Llama 3.1-70B on ALCUNA (Yin et al., 2023).

a new benchmark **ECHOQA** (Section 3.3) spanning diverse knowledge-intensive reasoning tasks, *i.e.,* scientific, factual and commonsense, across various LLMs. We extract PK by ensuring LLMs can reach 100% performance for the knowledge. Next, we construct CK from PK to fit our reasoning types, as in the middle of Figure 1. We hope that ECHOQA will serve as a valuable resource to steer future explorations on LLMs leverage and integration of PK with CK.

To investigate to what the extent LLMs can leverage PK, we design a series of reasoning instructions, which progressively enforce the use of PK, guiding LLMs preference to varying degrees, as detailed in Section 3.2. Furthermore, we investigate other factors affecting LLMs abilities to leverage their internal knowledge, *e.g.,* popularity of entities and methods of new knowledge fabrication, shedding lights on future improvements (Section 5). Our investigation draws the following findings:

• *The suppression of PK in the presence of CK is universal*, regardless of the model, knowledge type or reasoning type (Complementary, Conflicting or Irrelevant). LLMs *disregard* their own knowledge, *e.g.,* more than 60% of cases in scientific knowledge for all tested models, relying solely on the context for reasoning. This highlights the vulnerability of LLMs in leveraging PK.

• LLMs are more likely to recall their knowledge for some knowledge and reasoning types, *e.g.,* the commonsense knowledge. We find evidence indicating that the reason behind is likely the imbalance of knowledge in training corpus.

• *Explicit instruction can help LLMs remember more PK, but still way off from fully leveraging PK.* This implies more sophisticated prompt or framework design has the potential to solve this problem to a larger extent. We show some possible prompt templates in Appendix 17.

## 2 RELATED WORK

**Parametric Knowledge in LLMs** is encoded in models parameters through vast amounts of text data during pre-training. Previous research extensively explores how LLMs leverage PK for reasoning (Li et al., 2024; Yang et al., 2024b; Wang et al., 2024). Some reveal the challenges in leveraging PK (Wang et al., 2023a; Allen-Zhu & Li, 2023). These studies typically focus on models ability to utilize PK to perform various tasks, *e.g.,* multi-hop reasoning, without considering the influence of contextual knowledge (CK). While valuable, these analyses do not address how PK is dynamically applied in real-world contexts where CK also plays an important role. Moreover, PK embedded in language models can be outdated or incorrect, prone to hallucination (Elazar et al., 2021; Lazaridou et al., 2021; Zhong et al., 2023). Though some work propose to edit PK (Dai et al., 2022; Meng et al., 2022; 2023), additional effort would be needed for the loss of other abilities.

**Contextual Knowledge for LLMs** refers to the information (or tools) augmented in LLMs context window to enhance generation (RAG), which hopefully supplements LLMs internal PK to improve faithfulness (Guu et al., 2020; Qin et al., 2023; Mallen et al., 2023). Previous work primarily focuses on optimizing the retrieval of relevant documents for reasoning (Press et al., 2023; Asai et al., 2023; Zhuang et al., 2024; Gutiérrez et al., 2024). However, as documents contain diverse information, LLMs may be confused when reasoning, even with optimal retrieval, *e.g.,* when the given knowledge contradicts their knowledge (Xu et al., 2024). We find that even complementary CK can hinder

Table 1: Definition, example and metric of our reasoning types over various relationships between CK and PK. Note that we ensure PK is known by LLMs and not shown in context. $U_R, M_R, A_{ck}, A_{pk}$ denotes unknown ratio, memorization ratio, answers faithful to CK and PK, respectively.

| Reasoning Type | Example | Metric |
|---|---|---|
| *Supportive*
CK $\odot$ PK | $Q$: What's the shape of the earth? $A$: Spherical
CK : The earth is spherical.
PK : The earth is spherical. | $Acc, U_R$ |
| *Complementary*
CK $\wedge$ PK | $Q$: The shape of the planet Z-man was born on? $A$: Spherical
CK : Z-man was born on earth.
PK : The earth is spherical. | $Acc, U_R$ |
| *Conflicting*
CK $\oplus$ PK | $Q$: What's the shape of the earth? $A_{ck}$: Flat; $A_{pk}$: Spherical
CK : The earth is flat.
PK : The earth is spherical. | $M_R$ |
| *Irrelevant*
CK $\vee$ PK | $Q$: What's the shape of the earth? $A$: Spherical
CK : The earth orbits around the sun.
PK : The earth is spherical. | $Acc, U_R$ |

LLMs ability to fully leverage their knowledge. In this paper, we formulate the relationship between PK and CK to investigate the influence of different kinds of CK on LLMs utilization of PK.

**Interplay between Parametric and Contextual Knowledge** mainly includes behavioral analyses or fusion of both knowledge. Some assess models handling of new knowledge (Yin et al., 2023), focusing on knowledge construction. Other works elicit PK from LLMs and design frameworks to introduce conflict (Wang et al., 2023b; Xie et al., 2024) or irrelevant CK (Wu et al., 2024). However, the effectiveness of LLMs leveraging of PK under various CK is understudied. Besides, the focus on one type of knowledge may overlook the intricate relation between the two knowledge sources reflected real-world scenarios, *e.g.,* LLM-based search engines or chatbots like ChatGPT Plugins and New Bing (Nakano et al., 2021; Deng et al., 2023) or human reasoning process, *e.g.,* when exploring new environments, human intelligence can naturally integrate knowledge in brain with outside nature. While some studies explore the fusion of CK and PK for improved sufficiency (Jiang et al., 2023; Zhang et al., 2024), the intricate relationship between them is understudied. We aim to comprehensively examine LLMs ability to manage and integrate these two knowledge sources.

## 3 EXPERIMENTAL SETUP

As shown in Table 1, we design four reasoning types based on various relationships between PK and CK. Without the loss of generality, we adopt Question Answering (QA) as the reasoning task, following prior work (Mallen et al., 2023; Cheng et al., 2024). Formally, given an instruction, a question $Q$ and CK, an LLM leverages PK and CK to reason the answer $A$. As shown in Equation 1, if $Q$ is posed, then CK operated with PK under the reasoning types (*rt*) leads to the answer $A$. Note that for all settings, PK always holds true for Equation 1, meaning that we ensure the LLM has PK required for $Q$. In experiments, we vary CK based on our designed *rt* as shown in Figure 1. Different from Xie et al., 2024, we only present CK in the context, to assess the ability to utilize PK more naturally.

$$Q \rightarrow (\text{CK } rt \text{ PK} \rightarrow A) \quad \textit{where } rt \in \{\odot, \wedge, \oplus, \vee\} \textit{ and PK holds true.} \tag{1}$$

$\odot$ *Supportive* We have CK and PK convey the same information. Our prior experiments show that models behave consistently with or without CK (as expected), or LLMs can effectively adopt PK in this case. Therefore, we do not report this type in main results. (Refer to Appendix B.2 for statistics.)

$\wedge$ *Complementary* This type typically requires both PK and CK (multi-hop) for reasoning. We maintain knowledge sufficiency by assuring CK is new to LLMs and PK is known by the LLM. A key challenge is establishing meaningful connections between them. Previous study construct multi-hop questions by replacing entities with descriptions (Talmor & Berant, 2018; Huang et al., 2023), which does not ensure that CK is new to LLMs. To address this, we create *entity-profiles* that relate to PK. For the example in Table 1, given PK "*the earth is spherical*", CK can be "Z-man *was born on earth*", which is complementary to PK to answer "*The shape of the planet* Z-man *was born on*".

⊕ **Conflicting** In this type, a fact in CK is fabricated by *deletion, modification* or *extension* from PK. We can also create longer narratives around the fabricated fact. For the example in Figure 1, by modification, we can change "*Michael Jordan is a Basketball Player*" into "*Michael Jordan works as an ESPN basketball commentator for 20 years*". We ensure the question $Q$ require that exact fact to answer. We denote the answer that align with CK and PK as $A_{ck}$ and $A_{pk}$, respectively.

∨ **Irrelevant** For this type, we randomly sample irrelevant information as CK, under certain constraints, *e.g.,* CK for questions about occupation should still pertain to occupations. As the goal is to test LLMs ability to leverage PK, we only consider the case when PK is necessary for $Q$ in main results. We discuss LLMs ability to filter relevant knowledge in Appendix B.4.

## 3.1 EVALUATION METRICS

We design various metrics to assess LLMs ability to leverage PK, as shown in Table 1. Specifically:

• **Accuracy** ($Acc$) evaluates whether the output exactly matches the ground truth. For Supportive, Complementary, and Irrelevant reasoning types, each question $Q$ has a corresponding ground truth answer $A_{gt}$. We compute $Acc$ to assess model performance in these scenarios.

• **Memorization Ratio** ($M_R$) measures the extent to which the model adheres to PK for Conflicting Reasoning. It is calculated as $M_R = \frac{f_{pk}}{f_{pk}+f_{ck}}$, where $f_{pk}$ and $f_{ck}$ denotes the frequency of answer faithful to $A_{ck}$ and $A_{pk}$, respectively, following prior work (Longpre et al., 2021).

• **Unknown Ratio** ($U_R$) measures the ratio of models generating unknown. On one hand, $U_R$ reveals models uncertainty to some extent. On the other hand, since we make sure PK is required by $Q$, higher $U_R$ shows lower perception of PK. We calculate $U_R$ for all reasoning types.

## 3.2 REASONING INSTRUCTIONS

Recent post-training technique, *e.g.,* instruction-tuning, enables LLMs to fluently follow human instructions. To explore to what extent LLMs can leverage PK, we design a series of progressively-enforced instructions to guide models behaviors. Please refer to Table 17 for examples.

---
**Examples of Progressively-Enforced Reasoning Instructions**

*Neutral*: Combine the given information and your OWN knowledge to answer questions.

*Trust Yourself*: The given information is NOT SUFFICIENT, you should use your OWN knowledge combined with the information to answer questions.                    (Complementary)
If you think the given information is CONFLICTED with your knowledge, should USE your OWN knowledge to answer questions.                    (Conflicting)

*Speak Out Loud*: You must first OUTPUT your OWN knowledge about the question. If you think the given information is conflicted with your knowledge, should USE your OWN knowledge to answer questions.                    (Irrelevant)

---

• *Neutral Instruction* is the simplest prompt, where we treat LLMs as a helpful assistant with knowledge to answer questions. Note that we indeed ask LLMs to use their own knowledge.

• *Trust Yourself Instruction* proactively show the insufficiency or ask LLMs to trust themselves. Concerningly, such enforcing instruction is to test LLMs leverage of PK, but not practical in real-world scenarios where the sufficiency and trustworthiness of knowledge are not ensured.

• *Speak Out Loud Instruction* further pushes LLMs to explicitly utter its PK then trust themselves, mirroring the Chain-of-Thought reasoning (Wei et al., 2022). This differs from Xie et al., 2024, who separately construct PK and CK, and ask models to make a choice when PK and CK are both in context. Also, this enforcing instruction is not for practical use.

## 3.3 DATASET

We introduce our dataset **ECHOQA** to test how well LLMs echo their PK when encountering CK[2]. Questions are in multi-choice-QA format and provided an unknown option, following Xie et al.,

---
[2]Our dataset will be released for future research.

Table 2: Number of examples for each LLM for Complementary/Conflicting/Irrelevant reasoning type. The difference between LLMs is due to their different possession of knowledge.

| Models | ALCUNA (#) | ConflictQA (#) | MuSiQue (#) | OpenBookQA (#) |
|---|---|---|---|---|
| OpenAI o1 | 537/512/1,119 | 1,993 | 611 | 488 |
| GPT-4o | 631/740/1851 | 3,001 | 525 | 476 |
| GPT-4o-mini | 599/852/1,811 | 1,801 | 675 | 458 |
| Llama 3.1-70B | 1,090/550/1,809 | 2,372 | 635 | 397 |
| Llama 3.1-8B | 905/873/1,953 | 2,299 | 393 | 351 |
| Qwen 2-7B | 993/346/1,397 | 2,143 | 452 | 415 |

2024. The key step is to firstly obtain models PK by questioning about a knowledge triple, where a correct answer indicates that the LLM possesses the relevant knowledge, as in the left of Figure 1. Then, we construct CK from PK based on reasoning types, as in the middle of Figure 1. More details are in Appendix A. We conduct human evaluations on the quality of constructed CK and $Q$, showing that 97.6% and 95.3% of $Q$ and CK perfectly fit our reasoning types (Table 1). Notably, our construction can adapt to any knowledge-intensive task. Specifically, ECHOQA is adapted from the following datasets, covering scientific, factual and commonsense knowledge:

**ALCUNA** (Yin et al., 2023) is a scientific (biological) dataset featuring artifact entities. The knowledge is structured in json format, obtained by adding, variation and dropping from existing knowledge, from which we construct conflict CK. We adopt Knowledge Differentiation subset for Conflicting and Irrelevant Reasoning, and Knowledge Association subset for Complementary Reasoning, respectively. For fair comparison, we adopt facts under "variation" as conflict CK (Section 4.2) and further discuss how the way of CK construction affects LLMs awareness of PK (Section 5.2).

**ConflictQA** (Xie et al., 2024) provides factual parametric evidence elicited from LLMs, conflicting evidence and corresponding answers by an LLM-based framework. We adopt the PopQA subset (Mallen et al., 2023), questioning about a single fact. Complementary CK and questions are constructed by *entity-profiles* with templates (Table 5) for Complementary Reasoning. We adopt the original conflict knowledge and randomly-sample several irrelevant evidences with the same relation type for Conflicting and Irrelevant Reasoning, respectively.

**MuSiQue** (Trivedi et al., 2022) is a factual dataset featuring longer document and multi-step reasoning, with sub-questions and evidence. We firstly question LLMs with sub-questions to label supporting paragraphs as PK and CK. Then, we filter out examples with both CK and PK for Complementary Reasoning. We adopt Llama 3.1-70B (*zero-shot*) to construct conflict CK by modifying PK and generate related $Q$. Prompts are in Table 15. Other operations are similar to ConflictQA.

**OpenBookQA** (Mihaylov et al., 2018) is a commonsense multi-choice QA dataset. We elicit the commonsense PK by questioning LLMs with Chain-of-thought (Wei et al., 2022) and then filtering out ones with correct answers. We instruct Llama 3.1-70B with *one-shot* demonstration to fabricate CK and multi-hop $Q$ requiring both PK and CK for Complementary Reasoning. Please refer to Table 16 for prompt examples. Other operations are similar to MuSiQue.

### 3.4 LANGUAGE MODELS

We include multiple LLMs spanning diverse parameter scales, *i.e.,* two closed-source (OpenAI o1-preview (OpenAI, 2024), GPT-4o and GPT-4o-mini (OpenAI, 2023)) and three open-source (Llama 3.1-70B, Llama 3.1-8B (Meta, 2024) and Qwen 2-7B (Yang et al., 2024a)). These models have demonstrated strong performance in knowledge-intensive tasks and instruction-following capabilities, making them well-suited for our tasks. In ECHOQA, we filter PK for each LLM (as in the left of Figure 1) to assess their ability to integrate PK with ck, statistics of examples are detailed in Table 2. For all LLMs, we experiment with Chain-of-Thought (Wei et al., 2022) for behavioral analysis and adopt *zero-shot* setting to avoid the bias introduced by demonstrations. Other setups remain as default. Specific model API/checkpoints are listed in Appendix B.

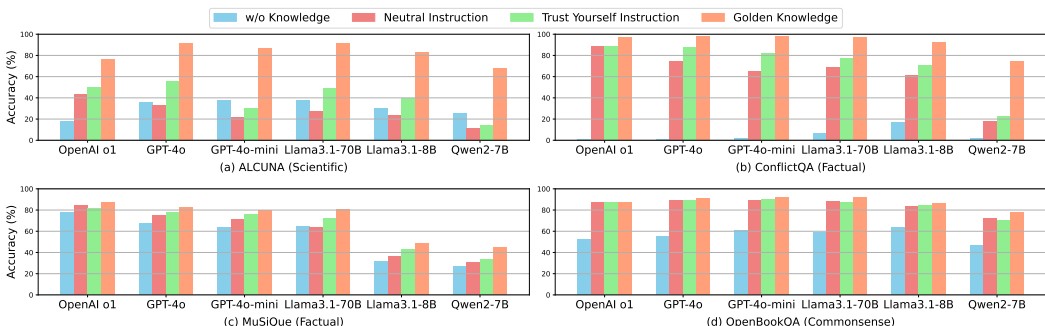

Figure 2: Accuracy for Complementary Reasoning. *w/o* Knowledge and Golden Knowledge means no information and all the required information is given, respectively. The upward trend shows that LLMs suppress PK even with complementary CK, comparing to the orange bar.

# 4 EXPERIMENTAL RESULTS

## 4.1 COMPLEMENTARY REASONING

**Although provided with complementary context, LLMs leverage of their own knowledge remains inhibited.** Figure 2 illustrates the Accuracy of LLMs for questions requiring both their parametric knowledge (PK) and the contextual knowledge (CK). Table 3 shows the Unknown Ratio $U_R$ on ALCUNA (more details in Appendix B.3). We also experiment with no knowledge and all required knowledge, *i.e.,* CK and elicited PK, in the context as the lower and upper bound (blue and orange bars), respectively. The following observations can be drawn:

First, *LLMs cannot fully leverage PK, even though the given CK with PK is sufficient for the question*. All tested LLMs across all knowledge types demonstrate an upward trend in Accuracy, with stronger models exhibiting higher absolute performance. With all required knowledge in context, LLMs perform effectively (orange bar). However, the performance drops dramatically with only CK in context (red/green bar). Figure 2 (a) (scientific knowledge) shows that the introduction of complementary CK even degrades the performance for most tested LLMs, compared to no knowledge given. We scrutinize 200 samples and find that the presence of CK hinders LLMs from engaging in deeper reasoning, *e.g.,* inferring entity properties by names, leading to overly rely on context. We further discuss this in Section 5.2.

Table 3: Unknown Ratio $U_R$ (%) for Complementary Reasoning on ALCUNA. *w/o* K, Gold K means none and golden knowledge is given, respectively. NI, TYI denotes Neutral and Trust Yourself Instruction, respectively. The **bold** denotes the highest $U_R$ in each row.

| Models | *w/o* K | NI | TYI | Gold K |
|---|---|---|---|---|
| *Closed-source LLMs* | | | | |
| OpenAI o1 | **65.78** | 46.12 | 35.43 | 20.63 |
| GPT-4o | 36.45 | **59.90** | 26.94 | 8.08 |
| GPT-4o-mini | 46.12 | **76.13** | 68.28 | 12.52 |
| *Open-source LLMs* | | | | |
| Llama 3.1-70B | 23.89 | **62.72** | 23.88 | 0.08 |
| Llama 3.1-8B | 18.12 | **58.50** | 34.35 | 12.70 |
| Qwen 2-7B | 40.28 | **81.26** | 73.62 | 28.60 |

Second, *with complementary CK, LLMs show even increased uncertainty.* Table 3 indicates that the complementary scientific CK "confuses" most tested LLMs significantly, doubling the $U_R$ for Llama and Qwen models, comparing "NI" with "*w/o* K" column. This implies that LLMs depend excessively on the relevant context to seek answers, suppressing their own PK even though PK is required for the question.

**Instruction-following aids in recalling some memories, but still way off.** By asking LLMs to adopt PK in a progressively enforced tone, our Reasoning Instruction (Section 3.2) significantly brings up the performance, demonstrating the strong instruction-following capabilities of LLMs and showing that instructions can modulate perception of knowledge to some extent. However, even when explicitly guided, LLMs still cannot fully recall the necessary PK they already have, leaving a large margin compared to when all knowledge is provided in the context. This highlights the need for dedicate prompt design for better integration of different sources of knowledge.

**Commonsense knowledge is easier to remember, while the scientific and factual is harder.** Comparing different types of knowledge in Figure 2, we have the following observations:

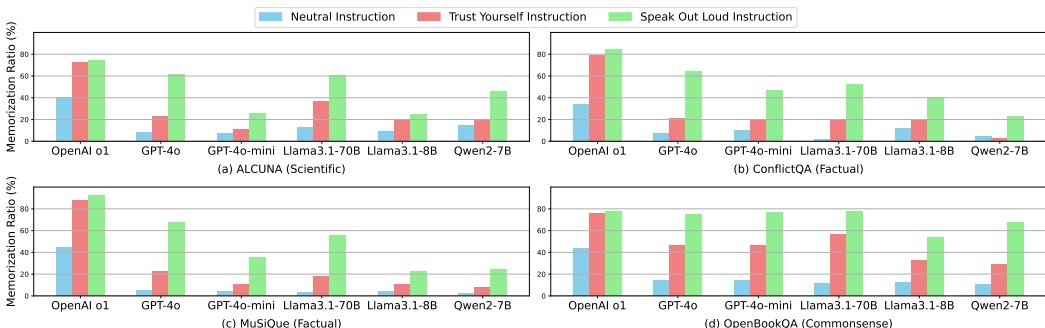

Figure 3: Memorization Ratio for Conflicting Reasoning. LLMs rarely trust themselves (PK) when faced with conflicting CK, though instructions modulate their preference to some extent.

First, LLMs exhibit nearly doubled accuracy in commonsense relative to scientific knowledge (red and green bar Figure 2 (a)(d)). We speculate the higher recall of commonsense PK is likely because commonsense information appears more frequently in the training corpus.

Second, LLMs can hardly answer without knowledge for ConflictQA (blue bar in Figure 2(b)). Different from other datasets, to fit the original question, we fabricate entity profiles and questions by templates in Table 5, *e.g.,* "*What's the occupation of the best friend of PersonA in high school?*", leaving little room for inference. This inversely suggests LLMs reasonable inferring ability.

Moreover, we observe that the upper bound for MuSiQue is considerably lower, especially for weaker models. However, the upward trend in performance persists, suggesting that while multi-step questions challenge LLMs reasoning ability, they may not significantly impair the recall of PK.

## 4.2 Conflicting Reasoning

**LLMs rarely trust themselves when faced with conflicting context.** Figure 3 depicts the ratio of LLMs relying on their parametric knowledge (PK) when presented with conflicting contextual knowledge (CK). Under Neutral Instruction, except for the strong OpenAI o1, few LLMs lean on themselves for more than 10% cases across all datasets. Note that the unknown ratio here is relatively low (more in Appendix B.3), making it reasonable to draw conclusions from Figure 3. It is demonstrated that *LLMs are faithful to the conflict context across all types of knowledge*, including commonsense, echoing findings in prior work (Xie et al., 2024). This observation underscores limitations in LLMs confidence and usage on their own knowledge when confronted with disinformation, warranting dedicated content filtering and system safety design for LLM-based systems.

**Instructions significantly influence LLMs preferences, but still struggled to fully recall their memory.** We further explore the extent to which LLMs can recall PK when faced with conflicting CK. As we instruct LLMs to trust themselves in a progressively more enforced tone, the Memorization Ratio ($M_R$) consistently goes up in all datasets, with stronger model exhibiting relatively higher $M_R$ (Figure 3). Compared to Neutral Instruction, the Trust Yourself Instruction almost doubled the $M_R$, with the Speak Out Loud Instruction further boosting the $M_R$. However, despite being instructed to trust themselves, few of $M_R$ exceed 60% for scientific and factual knowledge, indicating that LLMs still cannot fully recall their internal knowledge when the context is in conflict. Again, o1 exhibits more confidence than other LLMs, especially when told to "Trust Yourself", but there is still space for improvement. This again highlights the vulnerability of LLMs prone to disinformation.

**LLMs exhibit greater resistance to conflicts in commonsense knowledge.** Figure 3(d) shows that LLMs can recall commonsense knowledge significantly better, with our instructions leads to stronger enhancement than others. Specifically, most tested LLMs are confident against conflicting CK in over 60% cases with the "Speak Out Loud Instruction" (green bar). Since commonsense facts are less likely to become outdated or change over time, it is crucial that LLMs stand on themselves when encountering conflicting information. Although not perfect, this offers promising insights into preventing attacks. In contrast, while the upward trend still holds, most LLMs are much less likely to trust their own factual knowledge, often succumbing to conflicting CK (Figure 3(b)(c)), highlighting the importance of knowledge authenticity in LLM-based systems.

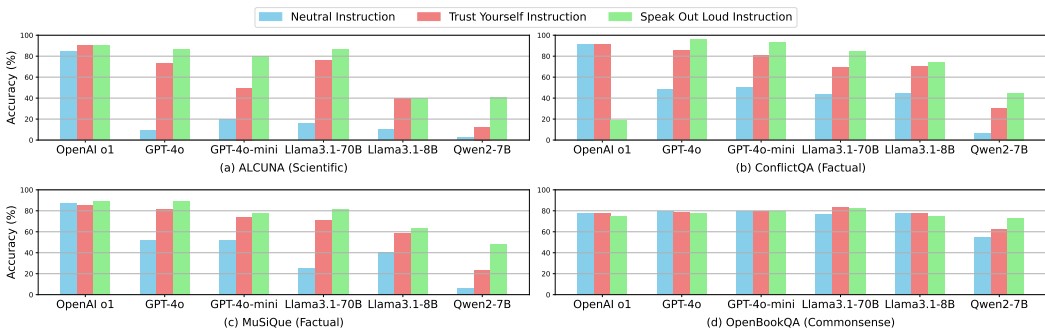

Figure 4: Accuracy for Irrelevant Reasoning, showing that LLMs relying on `CK` even though it is irrelevant and that instructions can substantially modulate their leverage of knowledge.

### 4.3 IRRELEVANT REASONING

**Although grasping the key to the knowledge, LLMs still seek answers in the irrelevant context.** Figure 4 illustrates the Accuracy for questions requiring LLMs knowledge (`PK`), when provided with one to four irrelevant documents in context (`CK`). Table 4 shows the Unknown Ratio ($U_R$) on ConflictQA (see Appendix B.3 for more details). Under Neutral Instruction, excepte for o1, most LLMs tend to search for answers solely within the `CK`, particularly in the case of scientific knowledge (blue bar), indicating *the difficulty LLMs face in disentangling `PK` from `CK` during reasoning*. Again, commonsense knowledge is more easily recalled, with GPT and Llama achieving up to 80% accuracy using only Neutral Instruction. We further discuss LLMs filtering ability in Appendix B.4).

We also observe that when all information is irrelevant, most LLMs exhibit high uncertainty, more than 50% Unknown Ratio for most tested models. With our Reasoning Instructions, the uncertainty decreases both gradually and substantially, dropping to below 5% for all tested models in ConflictQA (Table 4). Such result also suggests higher possibility of models leverage of `PK`. Moreover, the OpenAI o1 shows extraordinary ability echoing `PK` when `CK` is irrelevant, showing close results at different levels of instructions. This indicates the intricate design behind o1 system and sheds lights on future improvements.

Table 4: Unknown Ratio $U_R$ (%) for Irrelevant Reasoning on ConflictQA. NI, TYI, SOLI denotes Neutral, Trust Yourself and Speak Out Loud Instruction, respectively. The **bold** denotes highest $U_R$ in each row.

| Models | NI | TYI | SOLI |
|---|---|---|---|
| *Closed-source LLMs* | | | |
| OpenAI o1 | **6.12** | **6.12** | 0.98 |
| GPT-4o | **50.38** | 13.46 | 0.53 |
| GPT-4o-mini | **48.42** | 17.81 | 0.80 |
| *Open-source LLMs* | | | |
| Llama3.1-70B | **55.14** | 27.36 | 2.33 |
| Llama3.1-8B | **50.41** | 23.88 | 4.83 |
| Qwen2-7B | **80.59** | 41.82 | 4.57 |

**LLMs echo their knowledge better when the context is irrelevant.** Comparing the absolute performance with Conflicting and Complementary Reasoning, LLMs are able to better leverage `PK` for reasoning when `CK` is irrelevant. With our progressively-enforced Reasoning Instructions, LLMs further improve their memory recall by a large margin (Accuracy reaches 80% for most tested models Figure 4) and become more certain about their answers (Unknown Ratio drops drastically in Table 4). The observation, alongside the results for Complementary and Conflicting Reasoning (Figure 2, 3), indicates the possibility of leveraging `PK` for reasoning. However, *any knowledge presented in context tends to capture their attention to some extent.* The relationships between `PK` and `CK` greatly influence LLMs awareness and leverage of their `PK`.

## 5 DISCUSSION

### 5.1 MODELS LEVERAGE OF PARAMETRIC KNOWLEDGE GIVEN CONTEXTUAL KNOWLEDGE

In previous sections, we comprehensively investigate LLMs behavior in the dynamic integration of parametric knowledge (`PK`) and contextual knowledge (`CK`). Specifically:

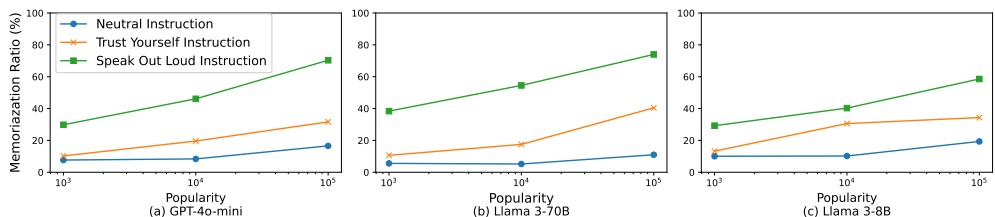

Figure 5: Memorization Ratio on ConflictQA across popularity categories, showing that LLMs recall their popular knowledge better.

• We identify four reasoning types based on various relationships between `PK` and `CK`. To test models ability to leverage `PK` when encountering `CK`, we introduce a new dataset ECHOQA, spanning scientific, factual and commonsense knowledge (Section 3).

• Our observations show that while LLMs possess the required knowledge, the presence of contextual information can significantly suppress their ability to leverage `PK`, regardless of the model, knowledge type, or reasoning type (Section 4).

• We also observed that certain types of knowledge are easier to recall. LLMs can better utilize `PK` for commonsense knowledge and when the context is irrelevant, compared to other types. This sheds lights on the improvement of integration of `PK` and `CK` to enhance reasoning performance.

• To improve the utilization of `PK`, we designed a range of reasoning instructions. Results demonstrate excellent instruction-following capabilities of LLMs. When instructed to "Speak Out Loud" or "Trust Yourself", LLMs substantially improve their memory recall. However, a significant gap still remains in fully leveraging `PK` when `CK` is present (Figures 2, 3, 4).

## 5.2 FACTORS AFFECTING MODELS LEVERAGE OF PARAMETRIC KNOWLEDGE

In previous findings, we show that the awareness and utilization of `PK` can be influenced by instructions, knowledge type, and the intricate relations with `CK`. Here, we discuss additional factors:

**LLMs can recall their popular knowledge more effectively.** Inspired by studies showing that LLMs lean on more popular entities, *i.e.,* monthly associated Wikipedia page views (Mallen et al., 2023; Xie et al., 2024), we evaluate representative LLMs on ConflictQA with Conflicting Reasoning *w.r.t,* popularity of the topic entity in the question, as illustrated in Figure 5. Intuitively, LLMs can recall memories better for more popular questions. This upward trend holds across all LLMs and our reasoning instructions. Specifically, GPT-4o-mini and Llama 3.1-70B achieve nearly 80% Memorization Ratio for the most popular questions, almost doubling that of the least popular ones. This observation might also explain why LLMs remember better for commonsense knowledge than others, indicating the reason behind is likely the knowledge frequency in training corpus.

**Introduction of knowledge relevant to the question may draw overly excessive attention to the context.** As mentioned in Section 4.1 (Figure 2(a)), counter-intuitively, even though the given `CK` with `PK` is sufficient to answer the question, performance can still degrade compared to direct questioning. We manually examine 200 random samples generated by Llama 3.1-70B with Neutral Instruction on ALCUNA where LLMs were able to answer on their own but failed when complementary information was introduced. Note that we do ask LLMs to use their own knowledge in the instruction (Section 3.2). We discover that in 93.5% of cases, LLMs relied on the context and answer choices and even claimed the insufficiency of knowledge (Refer to examples in Appendix B.5 and Table 14). This suggests that *LLMs tend to ignore their own knowledge, inhibit deeper thinking and depend entirely on the context*. Although our "Trust Yourself Instruction" improves performance, it is still far from expectation, indicating that *current LLMs struggle with integrating `PK` and `CK`*.

We speculate that the phenomenon may stem from the post-training process (Chung et al., 2024). LLMs are fine-tuned with instructions, question and optional context in a fashion shown in Equation 2. This training approach encourages LLMs to answer based either on the context (`CK`) or their internal knowledge (`PK`) separately, rather than integrating both. We hypothesize that this training objective is the reason behind LLMs struggles in effectively combining `PK` and `CK`.

$$\textit{Instruction; Context}(\textit{Optional}); \textit{Question} \rightarrow \textit{Answer} \qquad (2)$$

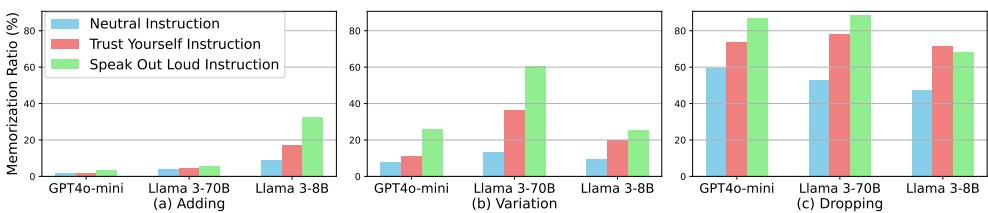

Figure 6: Memorization Ratio for Conflicting Reasoning under different ways of fabricating `CK` on ALCUNA. LLMs recall their knowledge way better with dropping than variation and adding.

**The method of new knowledge fabrication impacts LLMs recall of memory.** As mentioned in Section 3, conflicting facts (`CK`) can be introduced through various methods: *adding*, *variation*, and *dropping*. For fair comparison, we report results under "variation" in Figure 3, since all datasets introduce conflicts by default through this method. Here, we further investigate how different fabrication methods affects LLMs awareness and leverage of `PK`, by experimenting with ALCUNA in Conflicting Reasoning. Figure 6 provides the following insights:

First, *a significant gap exists between different fabrication methods*, although the overall trend observed in previous experiments remains consistent. This highlights that LLMs awareness of `PK` is strongly related to the how `CK` is fabricated from the original `PK`.

Second, *when new properties are added to an existing entity, LLMs are highly receptive to the new information*, exhibiting mostly lower than 10% Memorization Ratio. This raises safety concerns, as LLMs may be convinced of the addition of disinformation to known entities, *e.g.,* fake news.

Third, *dropping of known facts does less harms in LLMs memory recall*. Compared with the other methods, `CK` constructed by dropping provides some information about the entity irrelevant to the question. In Figure 6(c), with only Neutral Instruction, LLMs can leverage `PK` in up to 60% of cases, echoing results in irrelevant reasoning (Section 4.3) where LLMs recall their memories better than other reasoning types. However, they are still far from fully echoing their `PK`. We further discuss models ability to filter relevant information in Appendix B.4.

### 5.3 How to Better Leverage Parametric Knowledge Given the Context

As demonstrated above, LLMs struggle with integrating their parametric knowledge (`PK`) with contextual knowledge (`CK`). However, this ability is crucial for both artificial and human intelligence. We have designed instructions to elicit `PK` to some extent (Section 3.2). Although effective, these instructions are progressively enforced and may not be natural enough for practical application. The key challenge is to enable LLMs to naturally learn how to balance the two sources of knowledge. A promising direction is to develop an agent framework where LLMs assume a controlled amount of responsibility. By separating the process into two steps: first recalling the required knowledge, then reasoning, LLMs may avoid distraction from the context. Another potential solution is to fundamentally integrate `CK` and `PK` in post-training (Equation 2), which we leave for future work.

## 6 Conclusion

In this work, we comprehensively investigate how well LLMs can leverage their parametric knowledge (`PK`) in the presence of contextual knowledge (`CK`). We formulate four *reasoning types*: *Supportive, Complementary, Conflicting and Irrelevant*, based on diverse relationships between `CK` and `PK`. To facilitate the study, we introduce a new dataset ECHOQA spanning scientific, factual and commonsense knowledge for multiple LLMs. We design a series of *reasoning instructions* with progressively more enforced tones to showcase LLMs leverage of `PK`. We find that LLMs consistently struggle to fully leverage their `PK` when `CK` is present under different reasoning types. While the tailored reasoning instructions substantially improve their awareness of `PK`, they still fall short of expectations. Furthermore, we show that LLMs awareness of their `PK` can be affected by the relation between `CK` and `PK`, instruction, entity popularity, methods of fabricating `CK`, showing their vulnerability when faced with external knowledge. We hope that our work provides valuable resources and insights to guide future explorations in understanding LLM-based systems. We will release our code and dataset to facilitate future research.

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

## LIMITATIONS

Our study reals a key vulnerability in current LLMs: they struggle to fully leverage their parametric knowledge when contextual knowledge is present. However, there are some limitations to our work. First, we base our study primarily on contemporary LLMs with strong reasoning and instruction-following abilities. While we find evidence that this issue can be partially mitigated through dedicated prompt or framework design, we argue that the ability to effectively integrate PK with CK should be inherent to the base model. We aim for improvements in this area in future work. Second, while we consistently observe our findings across various reasoning types, we did not focus on making the data more challenging for broader coverage. For instance, conflicting knowledge could occur at the entity, sentence, or document level, and exploring such distinctions is left for future research. Third, real-world applications are likely more complex than the reasoning types we have designed. Although we categorize the interactions between PK and CK into different reasoning types, there may be more nuanced combinations of these relationships in real-world scenarios. We hope our findings raise awareness of the safety concerns surrounding the trustworthiness and reliability of LLM-based systems.

## APPENDIX

In the following sections, we detail on the follow aspects:

- Appendix A: Details of Data Constructions
- Appendix B: Details of Experimental Statistics
- Appendix C: Prompts Examples

## A    ELABORATION ON DATA CONSTRUCTION

As illustrated in Section 3.3, our proposed dataset ECHOQA aims to test LLMs awareness and leverage of parametric knowledge given contextual knowledge. EchoQA is adapted from existing datasets, the construction procedure of which can fit into any knowledge-intensive task. The key step is to obtain LLMs PK then construct CK and answer choices to fit our reasoning types. By default, the answer choice includes the answer sticking to CK and PK for Conflicting Reasoning or the ground truth for other reasoning types, a randomly-sampled option with the same type, and an unknown option. Note that we conduct human evaluation for all generated questions and knowledge to ensure the meaningfulness, knowledge faithfulness and fluency. Examples for each dataset is shown in Table 10, 11, 12, *w.r.t,* our reasoning types. Here we show more details.

**ALCUNA** (Yin et al., 2023) originally provides new knowledge. To obtain PK, we adopt a multi-choice question about the fact required by the original question. For example, *"Does cat prey on fish? A. Yes, B. No, C. Unknown"* or *"Cat prey on what? A. Fish, B. Tiger, C. Unknown"*. The exact fact for the question is provided by the dataset. The knowledge association subset naturally fits our Complementary Reasoning Type. Questions for other reasoning types are sampled from the knowledge differentiation subset. For Conflict Reasoning, we regard the new knowledge adapted from the old one as a conflict fact by changing the new entity name back. For example, given *"Dogcat prey on rabbit"*, we change it back to *"Cat prey on rabbit"* as a conflict fact. The entity name in the question also need to change accordingly. For Irrelevant Reasoning, we randomly sample 1-4 new entities profiles provided as CK.

**ConflictQA** (Xie et al., 2024) originally provides conflicting factual knowledge based on PopQA (Mallen et al., 2023) and StrategyQA (Geva et al., 2021). We adopt the PopQA subset, with the *popularity* of a question, counted by monthly Wikipedia page views associated with entities mentioned in the question. The key challenge is to construct new knowledge never seen by the LLMs for Complementary Reasoning. Following PopQA Mallen et al. (2023), we adopt templates for each relationship (Table 5). With the template, we obtain CK and construct new multi-hop questions. For example, for question *"What's the occupation of Michael Jordan"*, we adopt template to create a new entity "PersonA" with profile *"Michael Jordan is the famous friend of PersonA"*. And we change the question to *"What's the occupation of the famous friend of PersonA"*. For Irrelevant Reasoning, we randomly sample provided knowledge with the same relationship.

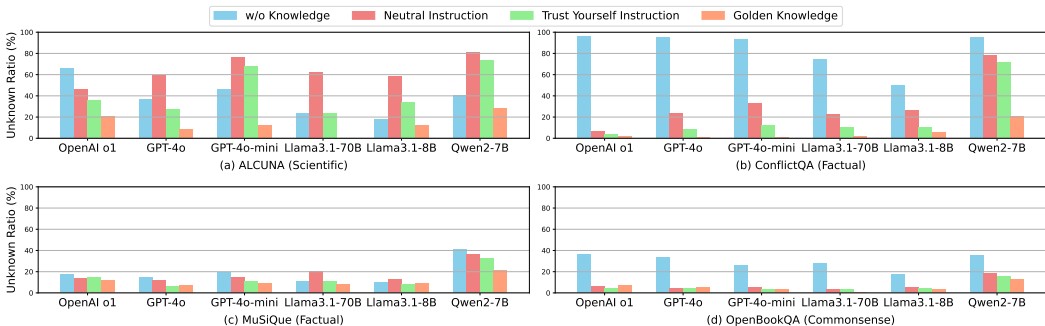

Figure 7: Unknown Ratio for Complementary Reasoning across four datasets. w/o Knowledge and Golden Knowledge means no information and all the required information is given, respectively. The $x$-axis denotes different models.

**MuSiQue** (Trivedi et al., 2022) provides all supporting paragraphs, from which we can query LLMs to obtain the `PK` and `CK` similar to ALCUNA. The sub-questions and documents can be directly used for Complementary Reasoning. We also adopt the answers of sub-questions as answer choices for Complementary Reasoning. The key challenge for this dataset is to create conflicting knowledge. We prompt Llama 3.1-70B by zero-shot to generate conflicting knowledge and a related question, based on the filtered `PK`, for Conflicting Reasoning. The prompt is shown in Table 15. Such questions are also used as for Irrelevant Reasoning, to test factual knowledge with longer context.

**OpenBookQA** (Mihaylov et al., 2018) provides some scientific facts. To obtain `CK` and `PK`, we query LLMs to answer the question in a Chain-of-Thought (Wei et al., 2022) manner (Refer to Table 16 for prompt examples), thereby we obtain the `PK` by filtering the correct answers. Then, we also prompt Llama 3.1-70B with one demonstration to generate complementary knowledge and questions (Refer to Table 16 for prompt examples). The Conflicting and Irrelevant Reasoning data construction is similar a MuSiQue.

## B ELABORATION ON EXPERIMENTS

### B.1 LANGUAGE MODEL API/CHECKPOINTS

This section elaborates on checkpoints we adopt for experiments. All open-source models are available on the Hugging Face platform. For Llama 3.1, we use "metallama/Llama-2.1-(8, 70)B-Instruct" respectively. For Qwen 2, we use "Qwen/Qwen2-7B-Instruct". We used OpenAI o1-preview, GPT-4o and GPT-4o-mini from OpenAI platform (https://platform.openai.com).

### B.2 RESULTS FOR SUPPORTIVE REASONING

Here we report the results for Supportive Reasoning on ALCUNA with representative LLMs. We do not include it in main experiments for two reasons: 1) Intuitively, models perform quite well. As we make share the `PK` is held by LLMs and `CK` has the same knowledge, the result is straightforward; 2) More importantly, it is hard to tell whether LLMs use `CK` or `PK` for reasoning when `CK` is present. We test questions require multiple and single facts rea-

Table 6: Accuracy for Supportive Reasoning, with multi-hop and single-hop questions on ALCUNA.

| Model | Multi-hop | Single-hop |
|---|---|---|
| GPT-4o-mini | 87.15 | 98.97 |
| Llama 3.1-70B | 91.30 | 97.80 |
| Llama 3.1-8B | 82.87 | 93.29 |

soning, as shown in Table 6. It is demonstrated that LLMs performance quite well with this type, reaching out expectation. However, based on our previous conclusions, LLMs might soly rely on context for reasoning, so they still may not perfectly leverage their own knowledge.

### B.3 ANALYSIS OF MODELS UNCERTAINTY

Here we report statistical results of Unknown Ratio $U_R$, which holds similar trend across all datasets and all tested LLMs. Therefore, we do not include all of them in the main body of our paper. Figure

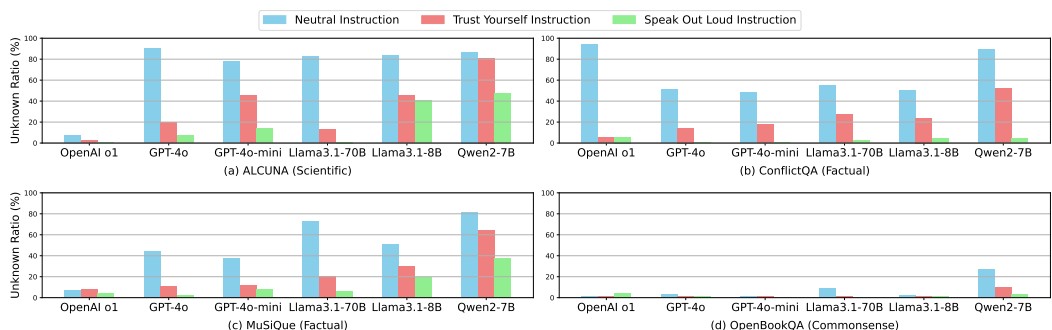

Figure 8: Unknown Ratio for Irrelevant Reasoning.

7 illustrates the $U_R$ for Complementary Reasoning. Similarly, Table 7, 8 and Figure 8 shows the $U_R$ for Conflicting and Irrelevant Reasoning, respectively.

Echoing discussion in the main body (Section 4.1), the introduction of complementary contextual knowledge may even bring up the uncertainty of models (Figure 7 (a)). With our progressively-enforced Reasoning Instructions, LLMs uncertainty go down substantially. But still, although given the sufficiency of knowledge, LLMs tend to choose "unknown" to some extent. This again echoes our finding that LLMs cannot fully leverage their knowledge when CK is present. Also, the $U_R$ for commonsense knowledge is quite low, compared with other types of knowledge, shedding lights on the future improvements.

For Conflicting Reasoning as shown in Table 7, 8, stronger models show relative lower uncertainty (lower than 7%). For weaker models in scientific knowledge, such as Llama 3.1-8B and Qwen 2-7B, while our progressively enforced instruction improves their leverage of PK and encouraging them to trust themselves, the $U_R$ also goes up. We speculate that while the absolute rate of trusting themselves increases, LLMs cannot well recall their memory of the scientific knowledge at the presence of CK, even when asked to output their own knowledge.

For Irrelevant Reasoning as shown in Figure 8, given the Neutral Instruction, LLMs tend to lean on context for reasoning, showing high uncertainty. When the context is irrelevant, although they have the knowledge to answer the question, they still tend to respond "Unknown", with Qwen 2-7B reaching 80% $U_R$. Concerningly, our progressively enforced instructions lower the uncertainty by a large margin, with "Speak Out Loud Instruction" almost dropping $U_R$ to lower than 10% for most LLMs. Also, LLMs exhibit less uncertainty when faced with commonsense knowledge, echoing previous conclusions that LLMs may recall this type of knowledge better than others.

### B.4 WHEN CK IS REQUIRED AND MIXED WITH IRRELEVANT INFORMATION

As our paper focuses on investigating LLMs ability to leverage PK, we do not include when PK is irrelevant to the question while the required CK is mixed with irrelevant knowledge. However, it is also an inevitable case for RAG-systems and shows LLMs ability of *knowledge discrimination*. We experiment on AL-CUNA with its Knowledge Understanding subset, to make sure the knowledge required for the question is not held by LLMs. Results are shown in Table 9. It is demonstrated that for knowledge new to LLMs, their

Table 9: Accuracy for Irrelevant Reasoning, requiring CK on ALCUNA. Irr Fact denotes randomly-sampled 1-4 irrelevant facts in context.

| Model | Single Fact | Irr Fact |
|---|---|---|
| GPT-4o-mini | 79.78 | 78.81 |
| Llama 3.1-70B | 86.81 | 87.75 |
| Llama 3.1-8B | 71.13 | 65.56 |

understanding ability is only decent, with Llama 3-8B reaching 71.13% Accuracy. However, with irrelevant documents, the result even goes higher, showing that the reasoning ability is not affected by the irrelevant knowledge. Compared to the main results in Figure 4, such result indicates that *LLMs can filter relevant documents very well in the context, but not disentangle their own knowledge if it is required for the question.*

## B.5 CASE STUDY

To figure out why LLMs suppress their own knowledge at the presence of contextual knowledge. We conduct case study over 200 random samples generated by Llama 3.1-70B with Neutral Instruction (Section 3.2) on ALCUNA (Yin et al., 2023) where LLMs were able to answer on their own but failed when complementary knowledge was introduced. Note that we do ask LLMs to use their own knowledge in the instruction (Section 3.2). We discover that in 93.5% of cases, LLMs solely rely on the context and even identified the inefficiency of knowledge for reasoning, as an example shown in Table 13. Without contextual knowledge, the LLMs can infer the new entity by its name, *e.g., dogcat might have properties similar to dog and cat.* Then, LLMs can reason by the process of elimination to figure out the correct answer. However, the introduce of CK is likely to limit LLMs reasoning to some extent in scientific knowledge. In other cases, the LLMs try to recall their own knowledge. But eventually distracted by the options or choose the wrong option by hallucination, as examples shown in Table 14.

## C PROMPT DESIGN

We provide a comprehensive list of all the prompts that have been utilized in this study, offering a clear reference for understanding our experimental approach.

Specifically, in Table 15 we provide prompt examples adopted to construct ECHOQA. In Table 17, we provide prompt examples for our designed Reasoning Instructions.

Table 5: Templates to connect new knowledge to parametric knowledge for PopQA. [subj] denotes subject entity in the new question. For example, for question *"What's the occupation of Michael Jordan"*, we adopt template to create a new entity with profile *"Michael Jordan is the famous friend of PersonA"*. And we change the question to *"What's the occupation of the famous friend of PersonA"*.

| Relationship | Template |
|---|---|
| Occupation | the person who frequently collaborates with [subj]
the best friend of [subj]
the famous friend of [subj] |
| Place of Birth | the renowned figure who is a friend of [subj]
the best friend of [subj]
the person who frequently collaborates with [subj] |
| Genre | the influential entity previously related with [subj] |
| Father | the renowned figure with [subj]
the famous person known by [subj]
the best friend of [subj] |
| Country | the renowned entity similar to [subj]
the notable figure similar to [subj] |
| Producer | the renowned product similar to [subj]
the famous product similar to [subj]
the prominent product like [subj] |
| Director | the renowned film similar to [subj]
the famous film similar to [subj]
the renowned movie similar to [subj] |
| Capital of | the famous place near [subj]
the renowned place near [subj]
the prominent place similar to [subj] |
| Screenwriter | the renowned screenplay similar to [subj]
the famous TV shows similar to [subj]
the prominent show like [subj] |
| Composer | the work similar to [subj]
the prominent work similar to [subj]
the influential work similar to [subj] |
| Color | the notable entity similar to [subj]
the prominent entity similar to [subj] |
| Religion | the best friend of [subj]
the prominent friend of [subj] |
| Sport | the prominent entity related to [subj]
the famous entity similar to [subj] |
| Author | the prominent work similar to [subj]
the influential work similar to [subj] |
| Mother | the renowned figure with [subj]
the famous person known by [subj]
the best friend of [subj] |
| Capital | the important place near [subj]
the big place near [subj]
the neighbor place of [subj] |

Table 7: Unknown ratio $U_R$ for Conflicting Reasoning on ALCUNA. NI, TYI, SOLI denotes Neutral, Trust Yourself, Speak Out Loud Instructions, respectively.

| Model | NI | TYI | SOLI |
|---|---|---|---|
| OpenAI o1 | 4.10 | 16.01 | 10.93 |
| GPT-4o | 6.22 | 3.10 | 6.48 |
| GPT-4o-mini | 2.34 | 1.76 | 1.76 |
| Llama 3.1-70B | 5.18 | 2.36 | 6.00 |
| Llama 3.1-8B | 10.99 | 13.51 | 30.58 |
| Qwen 2-7B | 21.38 | 16.47 | 19.36 |

Table 8: Unknown ratio $U_R$ for Conflicting Reasoning on ConflictQA. NI, TYI, SOLI denotes Neutral, Trust Yourself, Speak Out Loud Instructions, respectively.

| Model | NI | TYI | SOLI |
|---|---|---|---|
| OpenAI o1 | 7.29 | 5.13 | 3.49 |
| GPT-4o | 3.33 | 3.93 | 4.39 |
| GPT-4o-mini | 3.06 | 4.22 | 0.78 |
| Llama 3.1-70B | 1.98 | 1.99 | 3.02 |
| Llama 3.1-8B | 3.09 | 5.87 | 9.40 |
| Qwen 2-7B | 4.56 | 3.36 | 8.35 |

Table 10: Complementary Reasoning Examples in ECHOQA

**ALCUNA**
Question: What type of flowers are visited by an organism that is preyed upon by the co-roosting partners of 'Myotis lucifralis'?
0. Haustrum  1. Picea sitchensis  2. Scalariogyra  3. Stellaria media
Contextual Knowledge: Myotis lucifralis co-roost with Myotis nattereri ......
Ground Truth: 1

**ConflictQA**
Question: What is the person who frequently collaborates with PersonA's occupation?
0. cartoonist, illustrator  1. record producer  2. film director  3. unknown
Contextual Knowledge: Eleanor Davis is the person who frequently collaborates with PersonA.
Ground Truth: 0

**MuSiQue**
Question: What administrative territorial entity is the owner of Ciudad Deportiva located?
0. Tamaulipas  1. Nuevo Laredo  2. unknown
Contextual Knowledge: The Ciudad Deportiva ("Sports City") is a sports complex in Nuevo Laredo, Mexico....
Ground Truth: 0

**OpenbookQA**
Question: Which activity can help a person save money on lunch expenses if they usually eat lunch out?
Contextual Knowledge: DailyBite offers lunch coupons to frequent customers. FoodieClub is an expensive membership-based dining group...
Ground Truth: 0

Table 11: Conflicting Reasoning Examples in ECHOQA. $A_{ck}$ and $A_{pk}$ denotes answer sticking to contextual and parametric knowledge, respectively.

| **ALCUNA** |
|---|
| Question: What's the longitude of the location where Phcytodidae is commonly found? |
| 0. 1.61505 degrees   1. 93.05 degrees   2. unknown |
| Contextual Knowledge: Phcytodidae longitude 1.61505 degrees... |
| $A_{ck}$: 0, $A_{pk}$: 1 |

| **ConflictQA** |
|---|
| Question: Who was the screenwriter for The Hunt? |
| 0. Nick Cuse and Damon Lindelof  1.David Judah Simon  2.unknown |
| Contextual Knowledge: David Judah Simon is a highly acclaimed screenwriter ...... |
| $A_{ck}$: 1, $A_{pk}$: 0 |

| **MuSiQue** |
|---|
| Question: Was G̈reen̈recorded as a standalone album after the US tour in late 1977? 0. Yes  1. No  2. unknown |
| Contextual Knowledge: G̈reen̈was not recorded alone, but rather in conjunction with M̈otivation Radio:... |
| $A_{ck}$: 1, $A_{pk}$: 0 |

| **OpenbookQA** |
|---|
| Question: In what type of environments is fog more likely to form? |
| 0. Environments with higher humidity, such as marshes.  1. Environments with lower humidity, such as deserts.  2. unknown |
| Contextual Knowledge: Fog is more likely to form in dry environments with low humidity levels. |
| $A_{ck}$: 1 $A_{pk}$: 0 |

Table 12: Irrelevant Reasoning Examples in ECHOQA

| **ALCUNA** |
|---|
| Question: Does the longitude of Phcytodidae equal 1.61505 degrees? |
| 0. yes  1. no  2. unknown |
| Contextual Knowledge: Argasidae longitude 88.54616346104861 degrees... Lichida latitude 71.4882053945494 degrees ... |
| Ground Truth: 1 |

| **ConflictQA** |
|---|
| Question: Who is the author of The Program? |
| 0.Suzanne Young.  1. Gregg Hurwitz 2. unknown |
| Contextual Knowledge: Anna Todd is a bestselling author who gained popularity...Cormac McCarthy is a highly acclaimed American novelist and playwright... |
| Ground Truth: 0 |

| **MuSiQue** |
|---|
| Question: Where was the Philae lander operated from? |
| 0. DLR's Lander Control Center in Cologne, Germany  1. A secret facility in the United States  2. unknown |
| Contextual Knowledge: Empire Sports Network was an American regional sports network...Nova Sports is a premium Sports service in Greece and is owned by Forthnet Group... |
| 0. Eating lunch at DailyBite and using their coupons  1. Having lunch with FoodieClub  2. unknown |
| Ground Truth: 0 |

| **OpenbookQA** |
|---|
| Question: What is the primary source of sustenance for predators in an ecosystem? 0. Herbivores or other animals˜ 1. Plants and fung  2. unknown |
| Contextual Knowledge: Tree roots can push through obstacles like sidewalks as they grow... |
| Ground Truth: 0 |

Table 13: Examples where complementary knowledge drops the performance in ALCUNA. The LLMs reply excessively on context in these examples.

**Question:** What is the prey of the organism that shares a roosting place with the species 'Myotis lucifralis'?

0. Chara andina  1. Aldabrachelys  2. Noctuidae  3. Geomyidae

**Ground Truth**: 2

**Output w/o Knowledge**:

The species 'Myotis lucifugus' (little brown bat) shares a roosting place with 'Myotis lucifugus', so the correct species to consider is 'Myotis lucifugus'. Now, we need to determine the prey of 'Myotis lucifugus'. Bats are insectivores, and their diet consists of various insects. Among the given options, 'Noctuidae' is a family of moths, which are known to be preyed upon by bats.

Final Choice: 2

**Output w/ Knowledge:** The organism that shares a roosting place with the species 'Myotis lucifralis' is 'Myotis nattereri'. Since the provided information about 'Myotis lucifralis' does not include the prey of 'Myotis nattereri', but we know 'Myotis lucifralis' preys on 'Noctuidae', among others, and it is likely that 'Myotis nattereri' has a similar diet, but without the exact information about 'Myotis nattereri', the best answer is 'Unknown' since we can't be certain what 'Myotis nattereri' preys on. However, we can make an educated guess that it would be similar to 'Myotis lucifralis', but the options given do not include any prey of 'Myotis lucifralis' other than 'Noctuidae' which is a possible answer if we were certain they had the same prey. In this case, the best answer is 'Unknown' or 'Noctuidae' if we are guessing.

Final Choice: 4

**Question:** What is the species that Myotis lucifralis co-roosts with through three intermediate species?

0. Annona pruinosa  1. Plecotus auritus  2. Tuxedo drakei  3. Struvea thoracica

**Ground Truth**: 1

**Output w/o Knowledge**:

Myotis lucifugus, also known as the little brown bat, is a species of bat known to co-roost with other species. However, the question is specifically asking about co-roosting through three intermediate species. Given the options, I must consider which species is biologically plausible to co-roost with Myotis lucifugus through intermediate species. Plecotus auritus, also known as the brown long-eared bat, is known to co-roost with other bat species. Based on the information available, I consider it the most likely candidate among the options provided.

Final Choice: 1.

**Output w/ Knowledge:**

The given information about Myotis lucifralis only mentions that it co-roosts with Myotis nattereri. However, the question asks for a species that Myotis lucifralis co-roosts with through three intermediate species. Since there is no information provided about the co-roosting relationships of Myotis nattereri or any other potential intermediate species, we cannot determine the correct answer based on the given information. Therefore, the correct choice is "Unknown".

Final Choice: 4

Table 14: Examples where complementary knowledge drops the performance in ALCUNA. The LLMs inhibits deep thinking with contextual knowledge compared to ones without knowledge in these examples.

**Question**: What is eaten by the organism that is parasitized by 'Argasidae', and is itself a predator of another species? 0. Dasypus novemcinctus 1. Ochropleura candelisequa 2. Danaus plexippus 3. Anthurium fragrans 4. Unknown

**Contextual Knowledge:** Argasidae parasitize Homo sapienss......

**Ground Truth:** 0

**Output w/o Knowledge:** Argasidae is a family of ticks, commonly known as soft ticks. These ticks are ectoparasites that feed on the blood of mammals and birds. Given the options, we need to find an organism that is a predator of another species and can be parasitized by Argasidae. Dasypus novemcinctus, also known as the nine-banded armadillo, is a predator that feeds on various invertebrates and small vertebrates. It can be parasitized by ticks, including Argasidae. Ochropleura candelisequa is a moth, Danaus plexippus is a butterfly, and Anthurium fragrans is a plant, none of which are likely to be parasitized by Argasidae in the context of being a predator of another species. Final Choice: 0

**Output w/ complementary knowledge:** The question states that the organism is parasitized by 'Argasidae' and is itself a predator of another species. From the given information, we know that 'Argasidae' parasitizes 'Procyon lotor'. We also know that 'Procyon lotor' is eaten by 'Argasidae'. However, the question asks for what is eaten by the organism that is parasitized by 'Argasidae', and is itself a predator of another species, which in this case is 'Procyon lotor'. Since we don't have information about what 'Procyon lotor' eats in the provided data, we can't give a specific answer from the options. However, 'Procyon lotor' is known to eat various small animals including insects, among them, some species of moths such as 'Ochropleura candelisequa' cousins, which is close to option 1. Final Choice: 1

**Question:** Uredo has a host, which in turn serves as a host for another organism. What is the host of this second organism? 0. Anthaxia 1. Pelophylax ridibundus 2. Lolium perenne 3. Tradescantia 4.Unknown

**Contextual Knowledge:** Uredo have host Avena sativa......

**Ground Truth:** 2

**Output w/o Knowledge:** To answer this question, we need to break it down step by step. Uredo is a genus of rust fungi, which are known to be parasitic on plants. Given that Uredo has a host, which in turn serves as a host for another organism, we can infer that the host of Uredo is likely a plant. Considering the options provided, we should look for the one that is most likely to be a host for Uredo and then a host for another organism.

Looking at the options: - Anthaxia is a genus of jewel beetles, which are not typically hosts for rust fungi. - Pelophylax ridibundus is a species of frog, which is not a host for Uredo. - Lolium perenne is a species of ryegrass, a type of plant. - Tradescantia is a genus of plants commonly known as spiderworts.

Given that rust fungi typically infect plants, we can narrow down our options to Lolium perenne and Tradescantia. However, we need to consider which of these plants is more likely to be a host for another organism that also serves as a host for Uredo.

Considering the common hosts for Uredo and the possibility of another organism being hosted by the same plant, we should consider Lolium perenne. It is known that some rust fungi infect ryegrass, and this plant can also host other types of organisms.

Final Choice: 2

**Output w/ complementary knowledge:** The hosts of Uredo are listed in the provided information, which includes various plant species. Among the options given, we should look for a plant species that is known to be a host for another organism. Tradescantia is a plant genus that is known to be a host for various organisms, including fungi, insects, and other invertebrates. On the other hand, the other options do not directly relate to known hosts of Uredo or other organisms.

Final Choice: 3

Table 15: Prompt Examples to construct conflicting knowledge and questions for ECHOQA.

| Operation | Prompts |
|---|---|
| Conflict Knowledge and Question Generation | You are tasked with generating conflicting knowledge based on the given knowledge and creating a related multiple-choice question. Follow these steps: 1. Generate Conflict Knowledge: Create a conflicting statement or fact that contradicts the given knowledge. This conflict should challenge the established understanding. 2. Formulate a Multiple Choice Question: Create a question that asks for the truth regarding the conflicting knowledge. The question should be about the conflict knowledge. Provide two answer choices. Include: 0. One choice that aligns with the given knowledge. 1. One choice that supports the conflicting knowledge.

Your generated conflicting knowledge should contain information with similar volume as the given knowledge. The Label of choice should be 0. and 1. Do not output anything after the Desired Output. Here is the knowledge you should refer: [Taxon] Desired Output Format: Thought: [thought] Conflicting Knowledge: [new_knowledge] Question: [new_question_with_two_choice] Let's think step by step. |

Table 16: Prompt examples for OpenBookQA.

| Operation | Prompts |
|---|---|
| Commonsense Knowledge Elicitation | You are an expert with factual commonsense knowledge on a wide range of topics. You will be given a multiple-choice question requiring commonsense knowledge to answer. Your task is to: 1. Output the commonsense knowledge needed to answer the question, making it as concise as possible. 2. Output the answer based on the knowledge.

For the question and fact provided, provide the needed commonsense knowledge and the answer from the choices. Desired output format: Knowledge: ["knowledge1", "knowledge2", ...] Final Answer Choice: [Choice] DO NOT output anything after the Desired Output.

Question: Which of these would let the most heat travel through? Choices: A. a new pair of jeans. B. a steel spoon in a cafeteria. C. a cotton candy at a store. D. a calvin klein cotton hat.

Knowledge: ["Metal is a thermal conductor.", "Steel is made of metal.","Heat travels through a thermal conductor."]

Final Answer Choice: B

Question: [question_stem] Choices: [formatted_choices] Knowledge: |
| Complement Knowledge and Question Generation | You are an expert with factual commonsense knowledge on a wide range of topics.

Given a multiple-choice question, answer, and related commonsense knowledge, you must create a new question that requires both the given knowledge and new entities/knowledge to answer.

Step 1: Generate New Entities Create new entities representing concepts or objects related to each choice. These entities must be new and not directly mentioned in the original context. Step 2: Create New Knowledge Create one-hop facts connecting the fake entities to the original choice. Answering the question must require both new and original knowledge. The new knowledge should be a list of strings.

Desired Output Thought: [thought] New Question: [new_question] New Choices: [new_choices] New Knowledge: [new_knowledge] New Answer Key: [new_answer_key] Let's think step by step. |

Table 17: Prompt Examples for our designed various levels of Reasoning Instructions.

| | |
|---|---|
| w/o Knowledge | You are a powerful question-answering system.
Your task is to use your own commonsense knowledge to answer a multiple-choice question. If you are unable to answer the question, respond with the "Unknown" choice key.
Answer the following question with your choice (A, B, C, D, etc.): [Question]
Desired format: Thought process: [Thought process], Final Choice: [Final Choice].
Let's think step by step. |
| Neutral / Golden Knowledge Instruction | You are a powerful question-answering system.
Users will provide some information along with a question. Your task is to combine the information provided by the user with your knowledge to answer the question. If you are unable to answer the question, respond with the "Unknown" choice key.
Here is the basic information you can refer:[Taxon]
Answer the following question with your choice (A, B, C, etc.): [Question]
Desired format: Thought process: [Thought process], Final Choice: [Final Choice].
Let's think step by step. |
| Trust Yourself Instruction | You are a powerful question-answering system.
Users will provide some information along with a question. Your task is to combine the information provided by the user with your knowledge to answer the question.
If you are unable to answer the question, respond with the "Unknown" choice key.
Here is the basic information you can refer: [Taxon]
Answer the following question with your choice (A, B, C, etc.): [Question]
Note that the information above is not sufficient to answer the question, you should use your own commonsense knowledge to further reason the final answer.
Desired format: Thought process: [Thought process],
Final Choice: [Final Choice].
Let's think step by step. |
| Speak Out Loud Instruction | You are a powerful question-answering system.
Users will provide some information along with a question. Your task is to use the information provided by the user and your own knowledge to answer the question. If you are unable to answer the question, respond with the "unknown" choice.
Answer the following question with your choice (A, B, C, etc.): [Question]
You MUST first output your own knowledge about the question. I am sure you have the knowledge!
Here is the Given Knowledge: [Taxon]
Determine whether the given information about the question conflicts with your own knowledge.
If you think the Given Knowledge conflicts with your own knowledge, you MUST USE your own knowledge to answer the question.
Desired format: Thought process: [Thought process], My Own Knowledge: [My Own Knowledge], Final Choice: [Final Choice].
Let's think step by step. |

