# OpenReview forum: "Understanding the Interplay between Parametric and Contextual Knowledge for Large Language Models"
_ICLR.cc/2025/Conference — Submitted to ICLR 2025_

### Official Review · Reviewer_Nwuf · 2024-10-18

**Soundness:** 3
**Presentation:** 3
**Contribution:** 2
**Rating:** 6
**Confidence:** 4

**Summary:**

In this paper, the author proposes a new benchmark called `EchoQA`` to probe the subtle interactions between Parametric Knowledge (PK) and Contextual Knowledge (CK) in Large Language Models (LLMs). It considers the following four types of interactions: **Supportive, Complementary, Conflicting, and Irrelevant**. With this benchmark, the author aims to understand the behavior of LLMs in terms of the interaction between PK and CK, providing insights into existing models. Experimental results show that current models are more likely to rely on Contextual Knowledge rather than Parametric Knowledge even with dedicated instructions.

**Strengths:**

1.	It is good that the author introduces a new taxonomy to understand the interaction between PK and CK. This taxonomy—Supportive, Complementary, Conflicting, and Irrelevant—offers a useful framework for understanding these interactions.
2.	With the new benchmark, the author conducts a series of experiments to investigate the behavior of LLMs concerning PK and CK interactions. The results are interesting and provide valuable insights into existing models.
3.  The paper is well-structured and clearly written, making it easy to follow the author's arguments and findings.

**Weaknesses:**

1.	In the experiments, it would be better to discuss results from different types of models (e.g., DPO model, PPO model, SFT model, and so on). In the current version, the author only discusses the results from the base model (e.g. LLaMA-3.1-8B), which is not aligned with enhancing instruction-following abilities. With better instruction-following capabilities, LLMs may be able to follow instructions (e.g., *“Answer the question based on context/yourself”*) and effectively balance knowledge from PK and CK, rather than relying heavily on CK. Consider the author claims that the current models are more prone to rely on CK even with dedicated instructions (LINE 21-22), it would be interesting to see how different models (e.g. LLaMA-3.1-8B-Instruct) perform in this scenario.

**Questions:**

1.	In lines 84-85, the author mentions that the reason behind the behavior is the imbalance of knowledge in the training corpus. However, I am not sure how the author defines the “popularity” of knowledge in the training corpus. Does this refer to the frequency of the knowledge appears in the training corpus (e.g. CommonCrawl) of LLMs?
2. It would be good to discuss more relevant works on the interaction between LLMs and CK/PK.

---

> ### Author Response · Authors · 2024-11-20
> **Response to Reviewer Nwuf by Authors**
>
> Thanks for your insightful comments. We would like to clarify some points as follows:
>
> ## Comment 1: Discussion of Results from Different Types of Models
>
> >*In the experiments, it would be better to discuss results from different types of models (e.g., DPO model, PPO model, SFT model, and so on)...*
>
> Thank you for your insightful suggestions. Indeed, we adopted instruction-tuned models (e.g., **LLaMA-3.1-8B-Instruct**) as detailed in **Appendix B.1**, because they are better at following the designed prompts. However, the conclusions still hold for these models, as well as for closed-source models. We will make this clearer in the revision.
>
> Additionally, we indeed initially tried the base models before instruction fine-tuning (e.g., **LLaMA-3.1-8B**), but these models struggled to follow even simple instructions, making it difficult to extract meaningful outputs—even with few-shot demonstrations. This is why we did not include them in the paper.
>
> ---
>
> ## Comment 2: Clarification on "Popularity" of Knowledge in the Training Corpus
>
> The term ‘popularity’ is defined in the paper titled ["When Not to Trust Language Models: Investigating Effectiveness of Parametric and Non-Parametric Memories"](https://aclanthology.org/2023.acl-long.546/), which is provided in the ConflitQA (PopQA subset) dataset. It is measured by the number of monthly Wikipedia page views related to the entities mentioned in the question. In our study, based on the original paper, we also hypothesize that factual knowledge that is frequently discussed on the web tends to be more easily memorized by LLMs. We adopt this as a basis to analyze the potential reasons for how LLMs leverage PK.
>
> ---
>
> ## Comment 3: Discussion of Relevant Works on the Interaction Between LLMs and CK/PK
>
> Thank you for the suggestion. We acknowledge this gap and plan to include a more detailed discussion of relevant works on the interaction between LLMs and CK/PK in the revision.

---

> > ### Comment · Reviewer_Nwuf · 2024-11-23
> >
> > I would like to thank the authors for their detailed clarification.

---

> > > ### Author Response · Authors · 2024-11-24
> > > **Thanks to Reviewer Nwuf**
> > >
> > > Thank you very much for your reply. Please let us know if you have any other questions.

---

### Official Review · Reviewer_HQXc · 2024-11-03

**Soundness:** 3
**Presentation:** 3
**Contribution:** 3
**Rating:** 6
**Confidence:** 4

**Summary:**

This work investigates the interactions between an LLM's internal parametric knowledge and external contextual knowledge. It introduces a new benchmark, EchoQA, focusing on supportive, complementary, conflicting and irrelevant interaction types, to help inform these investigations.

**Strengths:**

The work is well-motivated and clearly presented. The experiments are well-structured and findings insightful. They are thorough and reveal various interesting and provoking questions about how LLMs operate across parametric and contextual knowledge. These include high level observations, such as that LLMs often disregard their internal knowledge in the presence of contextual knowledge, among a broader set of findings discussed in detail throughout the paper.

**Weaknesses:**

The work introduces a reasoning-oriented Question Answering task, but the Related Work section provides no mention of existing QA works designed to investigate model reasoning capabilities.

It is unclear which models are used to establish parametric knowledge (is this model-specific, or was one model selected as representative of all LLMs?). Similarly, further discussion of the effects of prompt templates to establish knowledge as presented in Table 5, in the main body of the paper seems valuable.

Deeper analysis of how different models perform across settings would also be insightful.

**Questions:**

Typos:
- L267: ck is not the right format

---

> ### Author Response · Authors · 2024-11-20
> **Response to Reviewer HQXc by Authors**
>
> Thank you for your insightful comments. We would like to clarify some points as follows:
>
> ## Comment 1: Lack of Mention of Existing QA Works in Related Work Section
>
> >*The work introduces a reasoning-oriented Question Answering task, but the Related Work section provides no mention of existing QA works designed to investigate model reasoning capabilities.*
>
> Thank you for your suggestion. We will include relevant related work regarding reasoning capabilities of LLMs in the revised manuscript to address this gap.
>
> ---
>
> ## Comment 2: Clarification on Parametric Knowledge (PK) Models and Prompt Templates
>
> >*It is unclear which models are used to establish parametric knowledge (PK). Is this model-specific, or was one model selected as representative of all LLMs? Similarly, further discussion of the effects of prompt templates to establish knowledge, as presented in Table 5, seems valuable.*
>
> We elicit PK for each model separately, providing both PK and CK for each model in each dataset in our **EchoQA** benchmark (refer to **Section 3.3**). This explains why there are various numbers of examples for each model in each dataset, as shown in **Table 2**.
>
> The consistent results observed across EchoQA suggest that the issues we highlight are not specific to a particular model, but rather generalize across the tested models.
>
> Thank you for the insightful suggestions on discussion of the effects of prompts templates. We did analysis the effect of our designed instructions in each sub-section of Section 4. We would love to clarify more on this in our revision.
>
> ---
>
> ## Comment 3: Deeper Analysis of Model Performance Across Settings
>
> >*Deeper analysis of how different models perform across settings would also be insightful.*
>
> Thank you for the valuable suggestion. Our current analysis focuses on showing that almost all LLMs tend to suppress their PK when CK is provided. We agree that a deeper analysis of model performance across different settings would be valuable. While different models do exhibit some variation in their performance, as discussed in each sub-section of **Section 4**, we plan to expand this analysis in the revision and dedicate a section to it.

---

> > ### Comment · Reviewer_HQXc · 2024-11-21
> >
> > I would like to thank the authors for their detailed clarification.

---

> > > ### Author Response · Authors · 2024-11-21
> > > **Official Comment to Reviewer HQXc by Authors**
> > >
> > > Thank you very much for your reply. Please let us know if you have any other questions.

---

### Official Review · Reviewer_HsAp · 2024-11-04

**Soundness:** 2
**Presentation:** 3
**Contribution:** 1
**Rating:** 3
**Confidence:** 5

**Summary:**

This paper explores how effectively large language models (LLMs) integrate internal parametric knowledge (PK) with external contextual knowledge (CK) to solve question answering tasks.
The relationship between PK and CK is categorized into four types: Supportive, Complementary, Conflicting, and Irrelevant.
 Based on these definitions, the paper synthesizes various types of evidence across four established QA benchmarks: ALCUNA, ConflictQ, MuSiQue, and OpenBookQA.
The synthesized evidence is then used to evaluate LLM behavior when presented with different types of CK, utilizing three distinct hand-crafted prompts.
Empirical results reveal a range of phenomena based on these evaluations.

**Strengths:**

a. This paper constructs a new dataset using their synthesized evidence of different types, which could be valuable for future research on the interplay between PK and CK.

b. The experiments demonstrate a considerable level of effort, including analyses involving four QA datasets from different domains and six different LLMs.

c. Some findings from this paper might be new and interesting, such as the significant variance observed when different prompts are used.

**Weaknesses:**

a. This paper’s contributions are modest, building upon prior work such as [1] and [2]. While it claims novelty through its evaluation of a broader range of relationships between PK and CK, including beyond just conflict, it is worth considering why conflicting knowledge has been the primary focus in the field—it is the most relevant and impactful aspect to study.  Notably, both the setup and evaluation in this paper are highly similar to those in [1], which was released over a year ago.

b. The paper would benefit from stronger focus and depth, as it currently presents a series of observations using various types of evidence without integrating them into a cohesive conclusion or offering actionable insights for the future design of RAG systems for LLMs. One potentially valuable direction that could have been explored in more detail is the sensitivity of LLMs to different types of prompts.

c. The experimental results in this paper highlight the significant impact of different prompt choices. However, the design of the three prompts used appears ad hoc and arbitrary. As a result, the empirical findings may not effectively translate to more practical, real-world scenarios.


A general issue with this paper is that the experimental setup could benefit from clearer motivation beyond simply aiming to differentiate itself from existing work. This raises questions about the overall value of examining relationships beyond knowledge conflict. Notably, the paper acknowledges that the supportive evidence scenario did not yield any surprising or meaningful results, so it was directly excluded from the main results. For the irrelevant evidence setup, the results appear to reflect primarily the influence of the ad hoc prompt. While the complementary evidence setup shows potential for more interesting insights, the paper’s lack of focus makes it challenging to draw in-depth conclusions from it.

[1] https://arxiv.org/pdf/2305.13300

[2] https://arxiv.org/pdf/2404.10198

**Questions:**

1. The numbers in Table 2 do not seem to be complete for ConflictQA/MuSiQue/OpenBookQA？

2. Why there's no speak out loud instruction in Figure 2?

3. The supportive type may not seem particularly interesting or surprising and was excluded from the main results for this reason. However, why do you still consider it important to include it in the final manuscript? Maybe just remove it from your paper.


Typos / Grammer issues:

line 72: LLMs leverage -> LLMs' leverage

line 812: reals -> reveals

---

> ### Author Response · Authors · 2024-11-20
> **Response to Reviewer HsAp by Authors -- Part One**
>
> We sincerely appreciate your detailed and thoughtful feedback. Below, we address the main points raised and aim to clarify any potential misunderstandings.
>
> ---
>
> ## Comment 1: Contribution of Paper Compared to Prior Work
>
> > *This paper’s contributions are modest, building upon prior work such as [1] and [2]...*
> > - **[1]** [https://arxiv.org/pdf/2305.13300](https://arxiv.org/pdf/2305.13300)
> > - **[2]** [https://arxiv.org/pdf/2404.10198](https://arxiv.org/pdf/2404.10198)
>
> While we acknowledge the contributions of [1] and [2], our work advances the field in the following ways:
>
> 1. **Distinct Setup**
>    Unlike [1] and [2], which focus on **LLM's preferences or bias** between PK and CK with a neutral instruction (primarily in conflicting cases), our methodology emphasizes LLMs’ **ability to truthfully utilize their parametric knowledge (PK)** across varying contextual knowledge (CK) types and instruction styles. In the instructions, we **explicitly ask the model to output their knowledge** regardless of the knowledge types. To the best of our knowledge, we are the first to discuss such important ability in literature.
>
>    For example:
>     - In **conflicting** scenarios, we enforce behavior by instructions like *“trust yourself”* to evaluate whether LLMs can echo their PK given the conflict CK. If CK states: “The world is flat” and the model is explicitly instructed to "trust yourself" and "speak out loud," the model should respond: “based on my knowledge, the Earth is spherical” instead of suppressing the knowledge. Such ability is important because failing to produce PK can lead to safety concerns, making the model prone to jailbreaking.
>     - In the **complementary** case (Figure 1), if the model knows "the occupation of Michael Jordan" and is given CK that “Person A’s best friend in high school is Michael Jordan”, it should be able to answer: “What’s the occupation of the person who is Person A’s best friend in high school”.
>
> 2. **Expanded Scope - Significance of Exploring Various Reasoning Types**
>    [1] and [2] focus on the construction of conflicting knowledge and analysis of LLMs behavior. Instead, we investigate relationships beyond conflict, including complementary and irrelevant scenarios.
>
>     The inclusion of multiple reasoning types (complementary, conflicting, irrelevant, supportive) is not only to broaden the scope but also to **provide a holistic investigation of LLM behavior**. Only based on conflict cases cannot fully reflect LLMs ability to echo their PK when CK is present. While conflict remains an important focus, our findings on complementary and irrelevant reasoning are also novel and impactful. We for the first time demonstrate the significant suppression of PK even when CK aligns and enhances PK (Section 4.1).
>
>     For example, when CK specifies "*Person A’s best friend is Michael Jordan*", LLMs often fail to use PK to infer *"Michael Jordan’s occupation"*, despite CK supporting it and its holding the PK.  This highlights critical insights into LLM behavior beyond conflict-focused studies.
>
> 3. **Evaluation and Results**
> The evaluations we adopted are based on the standard question answering metrics (similar to [1] and [2]). However, we highlight that almost ALL LLMs have difficulties in following instructions to truthfully output their internal knowledge, even when explicitly instructed to do so, regardless of reasoning types, models and knowledge categories (See **Figure 2, 3, 4*).
>
>    For example:
>     - In **complementary** scenarios, we show LLMs often fail to integrate CK and PK, even when both are necessary, highlighting limitations in reasoning beyond conflict.
>     - In **irrelevant** scenarios, we observe that LLMs often seek answers in the context even when they already know the answer and are explicitly asked to used their own knowledge when the context is irrelevant.
>
> Table R1 also summarizes some key differences:
>
> ### **Table R1**
>
> | **Aspect**             | **[1] and [2]**                             | **Our Work**                                                                                      |
> |-------------------------|---------------------------------------------|---------------------------------------------------------------------------------------------------|
> | **Focus**              | preferences on PK vs. CK  |  utilization of PK under varying CK types and instructions                                           |
> | **Instructions**       | Neutral or simple context prompts    | Explicit and progressively enforced instructions (*e.g., “trust yourself”*)                     |
> | **Reasoning Types**       | Conflicting    | Conflicting, Complementary, Irrelevant     |
> | **Ability Required**    |  Stantard question answering   | LLM should be able to use PK, either to combine PK with CK or "trust PK" to answer the question |

---

> ### Author Response · Authors · 2024-11-20
> **Response to Reviewer HsAp by Authors -- Part Two**
>
> ## Comment 2: Lack of Focus and Depth
>
> >*The paper would benefit from stronger focus and depth, as it currently presents a series of observations using various types of evidence without integrating them into a cohesive conclusion or offering actionable insights for the future design of RAG systems...*
>
> 1. **Focus of Our Paper**
> We comprehensively investigate to what extent LLMs can leverage their parametric knowledge (PK) when contextual knowledge (CK) is presented under various types of CK and prompts (see Section 1, Lines 47–49, and Section 3, Lines 148–149). We aim to show that almost ALL LLMs have difficulties in following instructions to truthfully output their internal knowledge, regardless of the relationships between CK and PK. This is emphasized throughout our paper.
>
> 2. **In-Depth Analysis**
>
> - **Tasks**: To facilitate deep analysis, we categorize the relationship between CK and PK into different reasoning types.
>
> - **Models**: We incorporate both strong closed-source models and open-source models with various parameter sizes.
>
> - **Instructions**: We cover a wide range of prompts, progressively and explicitly enforcing the model to leverage their own PK to answer questions.
>
> 3. **Consistent Conclusions**
> Our conclusions remain consistent regardless of models, reasoning types, and instructions, as summarized in **Section 1, Lines 78-88**.
>
> 4. **Actionable Insights for RAG Systems**
>
>     We respectfully clarify that:
>    -  The paper’s focus is not on designing RAG systems but on understanding how well LLMs can leverage PK when CK is present.
>    - While our findings may inform future RAG designs, our primary contribution lies in diagnosing vulnerabilities in LLM behavior.
>
>    That said, we do provide insights into RAG system design:
>     - **Section 4** identifies critical weaknesses in PK utilization.
>     - **Section 5.2** explains potential reasons for these weaknesses.
>     - **Section 5.3** offers insights into mitigating these issues, such as more effective prompting strategies.
>
>     These results align with the goal of analyzing LLM capabilities rather than proposing direct RAG improvements.
>
> ---
>
> ## Comment 3: Prompt Design
>
> >*The experimental results in this paper highlight the significant impact of different prompt choices. However, the design of the three prompts used appears ad hoc and arbitrary. As a result, the empirical findings may not effectively...*
>
> 1. **Why We Design Various Levels of Prompts**
> The focus of our paper is to understand LLMs' behavior under different prompts.
>
> 2. **How We Design Various Prompts**
> As detailed in **Section 3.2**, the prompts are progressively enforced to test LLMs' ability to utilize PK. While these prompts may not be natural for practical use (Lines 204-205, 208-209), their design targets LLMs' PK utilization. Examples are provided in **Table 17**.
>
> 3. **Results of Various Prompts**
> Our experiments demonstrate that LLMs' utilization of PK can be significantly influenced by the prompts, showing their effectiveness.
> However, our key point is that LLMs still cannot fully leverage their PK (Section 4 and Section 1 Line 86-88), instead of LLMs' being impacted by different prompts.
>
> ---
>
> ## Comment 4: Motivation and Value of the Study
>
> >*A general issue with this paper is that the experimental setup could benefit from clearer motivation beyond simply aiming to differentiate itself from existing work. This raises questions about the overall value of examining relationships...*
>
> The purpose of our paper is **NOT** to evaluate LLMs' bias or preference given CK and PK (where conflicting scenarios dominate prior works). Instead, we investigate to what extent the model is able to output PK when given various CK types under different prompts (**Section 1, Lines 47-49; Section 3, Lines 148-149**).
>
> Our use of varying levels of instruction tests whether LLMs can output PK when enforced to do so, a perspective distinct from [1], [2] and never explored in prior work .
>
> Please refer to **Comment 1** for details.
>
> ---
>
> ## Comment 5: Clarifications on Results of Supportive and Irrelevant Reasoning
>
> >*The paper acknowledges that the supportive evidence scenario did not yield any surprising or meaningful results, so it was directly excluded from the main results. For the irrelevant evidence setup, the results appear to reflect primarily the influence of the ad hoc prompt.*
>
> 1. **Results of Supportive Reasoning**
> We do not claim that these results are not meaningful. Instead, they are indeed showcased and discussed in **Table 6, Appendix B.2**. We don't cover it in the main body because when CK is supportive, it is hard to tell whether LLMs used their PK or not.
>
> 2. **Results of Irrelevant Reasoning**
> Although the results are influenced by instructions, the LLMs still cannot fully leverage their PK when CK is given as shown in Figure 4. The impact of instructions is analyzed in **Section 4.3, Lines 404-407**.

---

> ### Author Response · Authors · 2024-11-20
> **Response to Reviewer HsAp by Authors -- Part Three**
>
> ## Comment 6: Completeness of Table 2
>
> > *The numbers in Table 2 do not seem to be complete for ConflictQA/MuSiQue/OpenBookQA.*
>
> The numbers in Table 2 are **indeed complete**. For the ALCUNA dataset, there are three numbers for each model because this dataset provides subsets of different sizes suitable for Complementary, Conflicting, and Irrelevant reasoning types. For other datasets, the question counts are uniform across reasoning types.
>
> ---
>
> ## Comment 7: Absence of "Speak Out Loud Instruction" in Figure 2
>
> We did not include the results of the “speak out loud instruction” in Figure 2 just for better illustration. However, we can include these results in the revised manuscript. Below are the statistical results for some datasets, which are consistent with observations across other scenarios:
>
> ### ALCUNA
> | Acc (Unknown) | w/o Knowledge       | Neutral Instruction | Trust Yourself      | Speak Out Loud       | Golden Knowledge    |
> |---------------|---------------------|---------------------|---------------------|---------------------|---------------------|
> | **GPT-4o**       | 35.50 (36.45) | 33.44 (59.90)       | 55.78 (26.94)       | 51.94 (33.74)       | 91.97 (8.08)        |
> | **GPT-4o-mini**  | 38.05 (46.12) | 21.70 (76.13)       | 29.88 (68.28)       | 36.33 (58.57)       | 87.15 (12.52)       |
> | **Llama3.1-70B** | 37.62 (23.89) | 27.51 (62.72)       | 49.32 (23.88)       | 50.27 (11.38)       | 91.30 (0.08)        |
> | **Llama3.1-8B**  | 30.61 (18.12) | 23.61 (58.50)       | 39.53 (34.35)       | 33.37 (32.15)       | 82.87 (12.70)       |
> | **Qwen2-7B**     | 25.38 (40.28) | 11.08 (81.26)       | 14.00 (73.62)       | 28.97 (35.25)       | 68.18 (28.60)       |
>
> ### ConflictQA
> | Acc (Unknown) | w/o Knowledge       | Neutral Instruction | Trust Yourself      | Speak Out Loud       | Golden Knowledge    |
> |---------------|---------------------|---------------------|---------------------|---------------------|---------------------|
> | **GPT-4o**       | 0.93 (95.37)         | 74.54 (23.55)        | 87.84 (8.66)         | 91.13 (6.14)         | 98.20 (1.07)        |
> | **GPT-4o-mini**  | 1.90 (93.60)         | 64.71 (32.87)        | 82.53 (12.02)        | 79.50 (12.45)        | 97.75 (0.87)        |
> | **Llama3.1-70B** | 6.70 (75.00)         | 69.28 (22.42)        | 77.75 (10.30)        | 82.82 (5.92)         | 96.79 (1.87)        |
> | **Llama3.1-8B**  | 16.88 (50.02)        | 60.98 (26.92)        | 71.03 (10.48)        | 65.59 (9.80)         | 92.04 (5.79)        |
> | **Qwen2-7B**     | 1.64 (94.87)         | 18.01 (78.39)        | 22.54 (71.26)        | 31.17 (62.20)        | 74.94 (20.49)       |
>
> ### MuSiQue
> | Acc (Unknown) | w/o Knowledge       | Neutral Instruction | Trust Yourself      | Speak Out Loud       | Golden Knowledge    |
> |---------------|---------------------|---------------------|---------------------|---------------------|---------------------|
> | **GPT-4o**       | 67.24 (14.29)        | 75.24 (11.81)        | 78.10 (6.48)         | 76.76 (8.62)         | 82.10 (7.43)        |
> | **GPT-4o-mini**  | 63.70 (18.96)        | 71.56 (14.81)        | 76.00 (10.96)        | 75.84 (11.07)        | 79.41 (9.33)        |
> | **Llama3.1-70B** | 64.80 (11.20)        | 63.67 (20.45)        | 72.50 (10.60)        | 72.87 (9.68)         | 80.88 (7.72)        |
> | **Llama3.1-8B**  | 31.55 (9.83)         | 36.53 (12.26)        | 42.96 (7.89)         | 33.86 (8.25)         | 48.79 (8.86)        |
> | **Qwen2-7B**     | 27.21 (41.37)        | 30.53 (36.73)        | 34.29 (32.94)        | 30.75 (30.31)        | 45.13 (20.80)       |
>
> ---
>
> ## Comment 8: Relevance of the Supportive Reasoning Type
>
> > *The supportive type may not seem particularly interesting or surprising and was excluded from the main results for this reason. However, why do you still consider it important to include it in the final manuscript? Maybe just remove it...*
>
> In this paper, we investigate to what extent LLMs can leverage their PK given CK by categorizing the relationship between CK and PK into various reasoning types. For the **supportive type**:
> 1. It is **Practical in real-world applications** and thus important to consider.
> 2. The result is indeed shown and discussed in **Appendix B.2** to provide a complete picture of LLM behavior across all reasoning types.
>
> While it may not seem surprising, its inclusion ensures our study is comprehensive and applicable across various contexts.
>
> We hope these responses address the reviewers' comments and highlights the purpose, insights, and robustness of our work. Thank you for considering our clarifications.

---

> ### Author Response · Authors · 2024-11-25
> **We'll be glad if reviewer offers feedback for discussion in case there's any misunderstanding**
>
> Dear reviewer HsAp,
>
> We believe our reply has eased the proposed concerns. It might be helpful if you offer feedback, where further discussion can solve unnecessary misunderstandings. And we're glad to see if there's further discussion.
>
> Thank you very much!

---

> ### Author Response · Authors · 2024-11-26
> **Kindly request your reply**
>
> Dear Reviewer HsAp,
>
> Thank you for your valuable feedback and constructive comments. We kindly ask if you could let us know whether our responses address your concerns and adjust your overall rating accordingly.
>
> With only a few days remaining in the discussion period, we would greatly appreciate your engagement to ensure a constructive dialogue. If there are any remaining issues or clarifications needed, we would be happy to address them promptly.
>
> Sincerely,
>
> Authors of Submission 11963

---

> ### Author Response · Authors · 2024-11-29
> **Follow-up Response to Reviewer HsAp by Authors**
>
> Thank you for raising this point. We are happy to clarify the importance of the ability we discuss, its practical impact (e.g., RAG system), the purpose of instructions and motivations to address your concerns.
>
> ## **1. The Importance of the Ability We Study**
>
> Our paper identifies **an important vulnerability of LLMs** in knowledge-intensive tasks, which is *almost commonly-used LLMs have difficulties in following instructions to **truthfully output their internal knowledge***. Instead of proposing a new task, we categorize contextual knowledge (CK) and parametric knowledge (PK) relationships, which are common in real-world applications, to enable a comprehensive investigation. We believe such ability is important for the following reasons:
>
> ### **a. From the Perspective of Intelligence**
>
> In knowledge-intensive tasks, a truly intelligent system should effectively leverage both its intrinsic knowledge and any additional context to solve complex problems.
>
> For example, human intelligence can recall our own knowledge when presented with a question and context (as discussed in Section 1). Depending on the relationships between CK and PK:
>
> - **Supportive CK:** It can remind us of relevant knowledge.
> - **Complementary CK:** We can combine it with our knowledge.
> - **Conflicting CK:** we can still recall our knowledge relevant to the question **instead of suppresing it**.
> - **Irrelevant CK:** We can still effectively use our knowledge without being misled by the context.
>
> Our experiments ensure the model has the necessary knowledge to answer the question (**PK holds**). However, the introduction of CK often **suppresses the knowledge (PK)**, leading to situations where the model cannot recall or use its own knowledge.
>
> > For example, as shown in Figure 1, when the model knows (Michael Jordan, occupation, basketball player), and told (PersonA, best friend in high school, Michael Jordan), the model fails to answer "What's the occupation of the best friend of Person A in high school", which is counter-intuitive and firstly discussed in our paper.
>
> We find that this suppression occurs irrespective of the task or CK-PK relationship, **exposing a significant vulnerability and a gap in LLM intelligence for knowledge-intensive tasks**.
>
> ### **b. From the Perspective of Practical Use (RAG Systems)**
>
> We appreciate the reviewer's practical concerns and would like to elaborate.
>
> One intuitive application of our findings is in retrieval-augmented generation (RAG), which supplements LLMs with retrieved knowledge. While prior works study LLMs' preference given CK and PK or the performance of RAG, our paper focuses on whether LLMs can still leverage PK when augmented with CK with various enforced-level of instructions.
>
> **Key Observations (Section 4):**
>
> - **If LLMs effectively use PK (which is the ideal case):**
>   - In cases of complementary or irrelevant CK, LLMs can effectively use PK for reasoning.
>   - In conflicting cases, the users can decide between PK and CK (because LLMs are able to output PK).
>
> - **If LLMs fail to use PK (as observed in our paper):**
>   - The complementary or irrelevant CK suppresses PK, reducing reasoning capabilities and suggesting the need of highly precise retrieval methods.
>   - Conflicting CK raises **safety concerns**, as injected knowledge (by jailbreakers) can suppress PK, potentially leading to harmful outputs.
>
> ## **2. The Purpose of Various Levels of Instructions**
>
> We clarify that sophisticated prompt optimization is not our focus. The progressively-enforced instructions introduced are indeed **not intended for practical use**, as stated in Section 3.2 and previous response, but instead serve **to deepen our analysis**. Their purposes include:
>
> - **Studying the extent to which LLMs can recall their PK:**
>   - With "neutral" instructions, LLMs rarely recall PK.
>   - We aim to investigate with the strong instruction-following ability, can LLMs do so when being explictly told to.
> > We observe that the instructions truly brings up the recall of PK, but LLMs still cannot fully leverage their knowledge.
>
> - **Suggesting potential solutions (Section 5.3):**
>   - CK consistently influence LLMs recall of PK.
>   - The "speak out loud" instruction somewhat improves PK recall, indicating that an **agent-based framework**  could be a potential solution. This, however, is not our main focus and we would like to explore in the future work.
>
> ## **3. Our Motivation**
>
> Indeed, we have never stated that our motivation is *"to differentiate from existing work"*. Instead, please refer to **Comment 2, 4**, **1. The Importance of the Ability We Study** and **Section 1** for our motivation. We clarify **Comment 1** is to address the misunderstanding that our work *"builds on prior work"*. We hope this can help you realize the focus of our paper.
>
> We hope the clarifications address your concerns and **align with the criteria for an acceptable-level score**. Please let us know if further explanation is required!

---

> ### Author Response · Authors · 2024-11-30
> **Kindly request your reply,**
>
> Dear Reviewer HsAp,
>
> As the discussion period comes to a close, we sincerely hope to know whether our rebuttal has addressed your concerns. Ifit has, we wouldgreatly appreciate it if you could consider **raising our overall rating to an acceptance level**. If there are still any unresolved concerns, could you please let us know so we can address them to merit a higher score. Your guidance would be invaluable to us.
>
> We believe the review process aims to help improve our paper, and we deeply value your constructive feedback.
>
> Sincerely,
>
> Authors of Submission 11963

---

### Official Review · Reviewer_aSyk · 2024-11-04

**Soundness:** 2
**Presentation:** 3
**Contribution:** 2
**Rating:** 5
**Confidence:** 4

**Summary:**

This work investigate the interaction between the knowledge in model parameters (parametric knowledge, PK) and in-context knowledge (CK) by designing benchmark, ECHOQA, to explore if large language models could effectively utilized PK and CK together to solve the tasks. They define the PK by ensuring the LLMs could reach 100% performance for the knowledge and then define CK by construct them following four types of relation: supportive, complementary, conflicting, irrelevant. The experiments on ECHOQA seems to show that models are relying more on CK instead of utilizing PK (which might not be that surprising).

**Strengths:**

The direction about exploring the interaction between PK and CK is interesting.

The experiment and analysis seems comprehensive.

**Weaknesses:**

Though the direction is interesting, some designs might be further discussed. For example,
1. How to define PK? What types of knowledge are considered as parametric knowledge? In this work, they seem to focus on world (science, factual, commonsense) knowledge while this set might not be comprehensive and could be biased (for example, certain types of knowledge might appear more in the training set and thus less likely to be affected by in-context-knowledge and certain types of information is rare in pre-training corpus and might be easily modified by in-context-prompts).

2. Also, different models might have different sets of PK based on the training corpus. As a results, the benchmark might not be generalized well to different sets of models.

3. The knowledge in the context might be more desired to use in actual applications as well. In actual use cases, people will put their own/personalized knowledge in the context and ask questions, and thus naturally hope the model to rely on the in-context text. Especially for the conflicting part, i think it makes more sense for models to rely more on in-context knowledge. And I think it is natural if models are more relied on in-context-knowledge (they should receive higher attention scores if the knowledge is related to the questions).

4. There are also recent work on the interaction between PK and CK [1] where they encourage the model to utilize both in-context skills and built-in skills to solve the reasoning tasks (math and etc., Table 1, Table 25), which seems to make more sense compared to the conflicting part in the evaluation in this paper.  As in-context knowledge is limited, exploring the way to activate the abilities to utilize necessary PK in the model might make more sense.

[1] Skills-in-Context Prompting: Unlocking Compositionality in Large Language Models

**Questions:**

See weakness.

---

> ### Author Response · Authors · 2024-11-20
> **Response to Reviewer aSyk by Authors -- Part One**
>
> ## Comment 1: Definition and Scope of Parametric Knowledge (PK)
>
> > *How to define PK? What types of knowledge are considered as parametric knowledge? In this work, they seem to focus on world (science, factual, commonsense) knowledge while this set might not be comprehensive and could be biased ...*
>
> ### **Definition of PK**
> As mentioned in **Section 3.3 (Lines 229–230)**, parametric knowledge (PK) refers to factoid knowledge that can be represented as a knowledge triple, i.e., *(head entity, relation, tail entity)*. For example, *(Barack Obama, wife, Michelle Obama)*. PK is encoded in LLMs during pre-training and is critical for LLM-based systems such as chatbots (Refer to **Section 2, Lines 93–99**).
>
> ### **How PK is Obtained**
> As explained in **Section 1 (Lines 69–70)** and following previous work ([2](https://arxiv.org/pdf/2305.13300)), PK for each model is identified by ensuring 100% performance when queried directly. For instance:
> - Query: *“The wife of Barack Obama is __”*
> - Model Response: *“Michelle Obama”* (or its aliases)
>
> A correct response means the model possesses this knowledge.
>
> ### **Types of Knowledge Considered**
> Our focus is on **factoid knowledge**, which can be represented as knowledge triples. To ensure **both coverage and generalizability** of our insights, we investigate scientific, factual, and commonsense knowledge. These categories allow us to analyze how LLMs leverage PK when CK is introduced covering broad domains. Furthermore, the **frequency of knowledge in training data** is indeed discussed in **Section 5.2**.
>
> ### **Effect of Frequency in the Training Set**
> As discussed in **Sections 4 and 5.1**, our findings are consistent across all types of knowledge (see **Figures 2, 3, and 4**). In **Section 5.2 (Figure 5)**, we further analyze the effect of frequency in the training data. While frequency does have a moderate impact on how well LLMs leverage PK, it does not fully explain the performance gap. For example:
> - Even for frequently seen knowledge, models rarely exceed a **40% memorization ratio** under the "trust yourself" instruction (**Figure 5**).
>
> This suggests that LLMs struggle to fully leverage their PK, even for commonly seen facts, further suggesting the generalizability of our conclusions.
>
> ---
>
> ## Comment 2: Generalization of the Benchmark
>
> > *Different models might have different sets of PK based on the training corpus. As a result, the benchmark might not generalize well to different sets of models.*
>
> The purpose of our benchmark, **EchoQA**, is to assess LLMs’ ability to output PK when CK is present—an area not covered by existing datasets due to the lack of paired PK (encoded in LLMs) and CK (unknown to LLMs). The key contribution of EchoQA lies in providing this distinction: for each task, we for the first time identify the known knowledge (PK) and construct related unknown knowledge (CK) for commonly used LLMs (see Section 3.3).
>
> We provide such resources by eliciting PK for commonly used models separately and then constructing CK based on our designed reasoning types for each tested model in knowledge-intensive tasks (see **Section 3.3**). Our EchoQA does consider the differences in models’ training corpora, which is reflected by the varying numbers of examples per model in **Table 2**.
>
> The consistent trends observed across **Figures 2, 3, and 4** for various models and datasets validate the generalizability and value of our benchmark and our conclusions.

---

> ### Author Response · Authors · 2024-11-20
> **Response to Reviewer aSyk by Authors -- Part Two**
>
> ## Comment 3: Preference for Contextual Knowledge (CK)
>
> > *The knowledge in the context might be more desired to use in actual applications. In real-world cases, users will provide personalized context and naturally expect models to rely on it. For conflicting knowledge, it makes more sense for models to prioritize CK.*
>
> To clarify, the focus of this paper is **NOT** on LLMs’ preference or bias in choosing between CK and PK. Instead, we evaluate **how well LLMs can leverage their PK** under various types of CK and prompts (see **Section 1, Lines 47–49**, and **Section 3, Lines 148–149**). We show that almost ALL LLMs have difficulties in following instructions to truthfully output their internal knowledge, regardless of the relationships between CK and PK.
>
> We believe that this ability is critical for both human and artificial intelligence. In the instructions, we *explicitly enforce* the models to output their own knowledge to test such ability. For example:
>
> - In the **complementary case** (**Figure 1**), if the model knows *"the occupation of Michael Jordan"* and is given CK that *“Person A’s best friend in high school is Michael Jordan”*, it should be able to answer:
>   - *“What’s the occupation of the person who is Person A’s best friend in high school?”*
>
> - For **conflicting knowledge**, we agree that relying on CK often makes sense. However, we find that the models cannot output their own knowledge even when being explicitly asked to. Moreover, there are cases where failing to produce PK can lead to safety concerns, making the model **prone to jailbreaking**. For instance:
>   - If CK states: *“The world is flat”* and the model is explicitly instructed to *"trust your own knowledge if the context is conflicting or irrelevant"*, the model should be able to respond: *“based on my knowledge, the Earth is spherical”* instead of suppressing the knowledge.
>
> We will expand these examples and clarify these points further in the revision.
>
> ---
>
> ## Comment 4: Comparison to the mentioned work
>
> > *There are also recent works on the interaction between PK and CK [1] where they encourage the model to utilize both in-context skills and built-in skills to solve reasoning tasks (e.g., math, etc., Table 1, Table 25). This approach seems more practical than the conflicting evaluation in this paper.*
> > [1]: *Skills-in-Context Prompting: Unlocking Compositionality in Large Language Models*
>
> ### **Different Problem Definitions**
> Thank you for mentioning this work. Referring to **Comment 1** (Part One), our paper focuses on **factoid knowledge** that can be represented as a knowledge triple, e.g., *(Barack Obama, wife, Michelle Obama)*, specifically assessing how well LLMs can echo their parametric knowledge (PK) when contextual knowledge (CK) is present. We mainly discuss LLMs ability on knowledge-intensive tasks (See **Section 1 and Section 3.3**).
>
> In contrast, [1] addresses **compositional generalization of skills**, which involves **operations on objects or functions** (see **Section 1 and Figure 1 in [1]**).  For example, a function can transfer some words into a python list, requiring no factoid knowledge. They mainly discuss models’ ability to combine and apply **skills (functions) rather than leverage factual knowledge**, as we investigate in our paper.
>
> We will clarify the distinction between factoid knowledge and reasoning skills in the revised paper to address potential confusion.
>
> ### **Different Abilities Required**
> While both studies explore LLM capabilities, the **goals and tasks are fundamentally different**. [1] focuses on how to compose the provided skills (or functions) by in-context learning. However, in our paper, we aim to show that almost ALL LLMs have difficulties in following instructions to truthfully output their internal knowledge.
>
> ### **Different Ways of Study**
> [1] proposes SKiC prompts for better incorporate LLMs reasoning abilities (e.g., math reasoning) with the provided skills (or functions), aiming to improve LLMs performance on some reasoning tasks (which is not knowledge-intensive). However, we design various relationships between CK and PK, and various levels of instructions, aiming to comprehensively investigate to what extent LLMs are able to output their PK when CK is present.

---

> ### Author Response · Authors · 2024-11-25
> **We'll be glad if reviewer offers feedback for discussions in case there's any misunderstanding**
>
> Dear reviewer aSyk,
>
> We believe our reply has eased the proposed concerns. It might be helpful if you offer feedback, where further discussion can solve unnecessary misunderstandings. And we're glad to see if there's further discussion.
>
> Thank you very much!

---

> ### Author Response · Authors · 2024-11-26
> **Kindly request your reply**
>
> Dear Reviewer aSyk,
>
> Thank you for your valuable feedback and constructive comments. We kindly ask if you could let us know whether our responses address your concerns and adjust your overall rating accordingly.
>
> With only a few days remaining in the discussion period, we would greatly appreciate your engagement to ensure a constructive dialogue. If there are any remaining issues or clarifications needed, we would be happy to address them promptly.
>
> Sincerely,
>
> Authors of Submission 11963

---

> > ### Comment · Reviewer_aSyk · 2024-11-26
> >
> > Thanks for the detailed response! I will raise my score to 5.

---

> > > ### Author Response · Authors · 2024-11-27
> > > **Thanks to Reviewer aSyk**
> > >
> > > Dear Reviewer aSyk,
> > >
> > > Thank you very much for your time and efforts in reviewing our paper and responses. We believe your comments will enhance the quality of our paper. We sincerely appreciate your evaluations on our paper and the decision to raise the score.
> > >
> > > Respectfully, we remain open to further suggestions and wonder if you have any other concerns. We would be happy to address them promptly. Thank you again for your valuable feedback and for helping us improve our work.
> > >
> > > Sincerely,
> > >
> > > Authors of Submission 11963

---

### Official Review · Reviewer_3ETj · 2024-11-09

**Soundness:** 2
**Presentation:** 3
**Contribution:** 2
**Rating:** 5
**Confidence:** 4

**Summary:**

The paper analyzes interactions between LLM parametric knowledge and contextual knowledge. It finds that parametric knowledge often gets suppressed when contextual information is provided. The paper tests several prompt techniques to improve the recall of parametric knowledge. The authors also plan to release a benchmark that they used to test this phenomenon.

**Strengths:**

The paper sheds light on an important phenomenon that contextual information in some cases prevents LLMs from recalling parametric knowledge.

The authors tested several major LLMs with various reasoning type tasks and various instructions.

**Weaknesses:**

I ran examples provided in the paper on three LLMs (Llama3, ChatGPT, Gemini) and wasn't able to reproduce the problem/results.

More importantly, I'm not convinced that the problem definition is valid across all setups that the paper explored: if the prompt provides distracting or conflicting knowledge, why should we expect the models to generate parametric knowledge and not follow the prompt?

This paper does not propose a solution, only highlights the problem.

The EchoQA dataset that will accompany this paper is a reformatted and processed version of four previously released datasets. It is not clear how much effort/novelty went into compiling this dataset.

**Questions:**

Is it even correct to expect from LLMs to rely on parametric knowledge recall (rather than following a prompt) when the prompt contains distracting/conflicting information?

---

> ### Author Response · Authors · 2024-11-20
> **Response to Reviewer 3ETj by Authors**
>
> Thanks for your insightful comments. We would like to clarify some points as follows:
>
> ## Comment 1: Reproducibility of Examples
>
> > *I ran examples provided in the paper on three LLMs (Llama3, ChatGPT, Gemini) and wasn't able to reproduce ...*
>
> Thank you for your observation. We would like to clarify that the examples in the appendix are provided to illustrate the data format across the datasets, not to serve as universal test cases. Since each LLM is trained on distinct datasets, their internal parametric knowledge varies. For evaluation, we assess whether the LLM can output the answer or A_pk given the question, contextual knowledge (CK), and instructions.
>
> To be more specific, the examples provided in the appendix are specific to Llama 3.1-8B-Instruct and may not yield equivalent results with other models. We will explicitly mention this in the revised paper to avoid confusion.
>
> ---
>
> ## Comment 2: Validity of the Problem Definition
>
> > *I'm not convinced that the problem definition is valid across all setups that the paper explored: if the prompt provides distracting or conflicting knowledge, why should we expect the models to generate parametric knowledge...*
>
> ### **Problem Definition**
> Thank you for raising this important point. To clarify, the focus of this paper is **NOT** on the preference or bias of LLMs in choosing between parametric knowledge (PK) and contextual knowledge (CK). Instead, we investigate the extent to which LLMs can leverage their PK under various types of CK and prompts (see Section 1, Lines 47–49, and Section 3, Lines 148–149). We aim to show that **almost ALL LLMs have difficulties in following instructions to truthfully output their internal knowledge**, regardless of the relationships between CK and PK.
>
> We believe this ability is critical for both human and artificial intelligence. For example:
> - As shown in **Figure 1**, if an LLM knows “the occupation of *Michael Jordan*” and is provided with CK that “Person A’ s best friend in high school is *Michael Jordan*”, the LLM should be able to answer:
>   - *“What is the occupation of the person who is Person A’s best friend in high school?”*
> - Similarly, given conflict/distracting knowledge, e.g., *"the earth is flat"*, and asked to *"trust your own knowledge if CK is conflicting/distracting"*, the LLM should be able to output its knowledge:
>   - *"based on my knowledge, the earth is spherical"*.
> Such ability is crucial concerning the safety situations.
>
> ### **Instructions**
> To test the ability, we **explicitly enforce** behavior in the instructions:
> 1. **"The CK is not sufficient, combine PK with CK"** when CK is complementary.
> 2. **"Trust your own knowledge / Output your own knowledge about the question"** when CK is conflicting or irrelevant.
>
> As discussed in Section 3.2, we instruct the model accordingly. Therefore, ideally, if the LLM possesses the PK, it should demonstrate the ability to utilize this knowledge, regardless of the CK type.
> We will elaborate further on these instructions and clarify the problem definition in the revision.
>
> ---
>
> ## Comment 3: Scope of the Paper
>
> > *This paper does not propose a solution ...*
>
> We appreciate this comment. Our paper is designed to comprehensively investigate the LLMs’ ability to leverage PK when CK is present, through:
> - Rigorous problem definition.
> - Experimental design and data construction.
> - Analysis of various reasoning types, models, and knowledge categories.
>
> While the primary focus is to analyze and uncover challenges, we also:
> 1. Provide **insights into the underlying reasons** for these challenges (Refer to **Section 5.2**).
> 2. Discuss **potential solutions** and highlight upward trends observed across prompts (Refer to **Section 5.3**), suggesting directions for future improvements.
>
> We will emphasize these points more clearly in the revised paper.
>
> ---
>
> ## Comment 4: Novelty and Efforts in EchoQA Dataset
>
> ### **Novelty**
> As mentioned in **Section 1**, the focus of this work is to examine LLMs’ ability to handle knowledge-intensive tasks. To this end, we utilize existing datasets that require various kinds of knowledge (scientific, factual, and commonsense). However, these datasets do not specify whether the knowledge is known or unknown to any model.
> - The key contribution of EchoQA lies in providing this distinction: for each task, we identify the **known knowledge (PK)** and construct **unknown knowledge (CK)** for commonly used LLMs (see **Section 3.3**).
>
> ### **Efforts**
> We provide PK and CK for each LLM across all datasets. This involves:
> - Extracting PK for each model.
> - Designing CK based on our proposed reasoning types (Refer to **Section 3.3**).
>
> For example, in MuSiQue, we first identify PK for each model and then generate **conflicting factual knowledge** and formulate corresponding questions to assess the model’s ability. We will further clarify this distinction and highlight our contributions in the dataset construction process.

---

> > ### Comment · Reviewer_3ETj · 2024-11-21
> > **Thank you for responses**
> >
> > Thank you to the authors, I really appreciated detailed responses! I'll keep my score, as I still think that the work is a bit narrow for ICLR.

---

> ### Author Response · Authors · 2024-11-21
> **Thank Reviewer 3ETj for responses - clarifications of our contributions**
>
> Thank you very much for your timely reply. We would love to clarify more about our contributions in case there is any other confusion.
>
> 1. **Motivations**
>
> As RAG-based systems (e.g., chatbots and new-bing) are widely used, we try to analyze the vulnerabilities of LLMs for future development. We comprehensively investigate to what extent LLMs can leverage their parametric knowledge (PK) when contextual knowledge (CK) is presented. (Section 1)
>
> 2. **Experiment Designs**
>
> For a comprehensive investigation, we study under various **types of reasoning based on different relationships between PK and CK, categories of knowledge (i.e., scientific, factual and commensense), models and instructions** (see Section 1, Lines 47–49, and Section 3, Lines 148–149). And for the first time, we identify the PK and related CK for knowledge-intensive tasks aross scientific, factual and commonsense knowledge for various commonly-used LLMs, and we will publish the resources. (Section 3)
>
> 3. **Insightful and Consistent Conclusions**
>
> The experiments show that **almost ALL LLMs have difficulties in following instructions to truthfully output their internal knowledge**, regardless of the relationships between CK and PK, the models and reasoning types. (Section 1 and Section 4)
>
> We believe that such vulnerabilities are critical for LLM-based systems (e.g., RAG systems). In addition, human intelligence can fluently echo our knowledge when presented some contexts (Section 1). We also provides some insights for the reasons of this problem and discuss how to mitigate the problem in Section 5.
>
> We hope this response ease your concerns effectively. If you have further questions, we would be happy to discuss them in more detail.

---

> ### Author Response · Authors · 2024-11-26
> **Kindly request your reply**
>
> Dear Reviewer 3ETj,
>
> Thank you for your valuable feedback and constructive comments. We kindly ask if you could let us know whether our responses address your concerns and adjust your overall rating accordingly.
>
> With only a few days remaining in the discussion period, we would greatly appreciate your engagement to ensure a constructive dialogue. If there are any remaining issues or clarifications needed, we would be happy to address them promptly.
>
> Sincerely,
>
> Authors of Submission 11963

---

> > ### Comment · Reviewer_3ETj · 2024-11-26
> > **thank you again**
> >
> > Thank you for re-iterating the contributions to me, although I understood them before as well. As I responded earlier, I believe that the contributions are a bit too narrow for a solid ICLR paper, so I will keep my score at 5.

---

### Meta-Review · Area_Chair_tJ5T · 2024-12-25

**Metareview:**

This paper investigates the interactions between parametric knowledge (PK) and contextual knowledge (CK) in large language models (LLMs) through the introduction of EchoQA, a benchmark that categorizes PK-CK relationships into four types: Supportive, Complementary, Conflicting, and Irrelevant. The findings reveal that LLMs often over-rely on CK, suppressing PK even when CK is complementary or irrelevant, and struggle to fully leverage PK despite tailored prompts.

Strength: The problem of LLM resolving conflicting information between CK and PK is important, especially in RAG settings. The paper introduces a novel benchmark, EchoQA, to explore PK and CK interactions, providing a framework for understanding these dynamics. It reveals insights such as the tendency of LLMs to suppress internal knowledge in favor of contextual inputs.

Weakness: One main weakness of the paper is that the findings that LLMs tend to suppress PK and in favor of CK is a relatively known problem. Given that the problem has been known but no solution or additional insights are provided, the significance of the paper contribution is relatively limited. On other weakness raised by reviewers is the generalizable of the work to other models is questionable. One reviewer questioned the reproducibility of the results using the prompt on other models.

**Additional Comments On Reviewer Discussion:**

Some concerns and questions have been addressed during the rebuttal. However, the main weakness mentioned above (lack of sufficient contribution) still remains.

---

### Decision · Program_Chairs · 2025-01-22

Reject